# InstaFlow: One Step is Enough for High-Quality Diffusion-Based Text-to-Image Generation

**Xingchao Liu**[1*], **Xiwen Zhang**[2], **Jianzhu Ma**[2], **Jian Peng**[2], **Qiang Liu**[1]

[1] Department of Computer Science, University of Texas at Austin

[2] Helixon Research

`xcliu@cs.utexas.edu, xiwen@helixon.com`
`majianzhu@tsinghua.edu.cn, jianpeng@illinois.edu, lqiang@cs.utexas.edu`

## Abstract

Diffusion models have revolutionized text-to-image generation with its exceptional quality and creativity. However, its multi-step sampling process is known to be slow, often requiring tens of inference steps to obtain satisfactory results. Previous attempts to improve its sampling speed and reduce computational costs through distillation have been unsuccessful in achieving a functional one-step model. In this paper, we explore a recent method called Rectified Flow [45; 43], which, thus far, has only been applied to small datasets. The core of Rectified Flow lies in its *reflow* procedure, which straightens the trajectories of probability flows, refines the coupling between noises and images, and facilitates the distillation process with student models. We propose a novel text-conditioned pipeline to turn Stable Diffusion (SD) into an ultra-fast one-step model, in which we find reflow plays a critical role in improving the assignment between noises and images. Leveraging our new pipeline, we create, to the best of our knowledge, the first one-step diffusion-based text-to-image generator with SD-level image quality, achieving an FID (Fréchet Inception Distance) of 23.3 on MS COCO 2017-5k, surpassing the previous state-of-the-art technique, progressive distillation [58], by a significant margin ($37.2 \rightarrow 23.3$ in FID). By utilizing an expanded network with 1.7B parameters, we further improve the FID to 22.4. We call our one-step models *InstaFlow*. On MS COCO 2014-30k, InstaFlow yields an FID of 13.1 in just 0.09 second, the best in $\leq 0.1$ second regime, outperforming the recent StyleGAN-T [73] (13.9 in 0.1 second). Notably, the training of InstaFlow only costs 199 A100 GPU days. Codes and pre-trained models are available at `github.com/gnobitab/InstaFlow`.

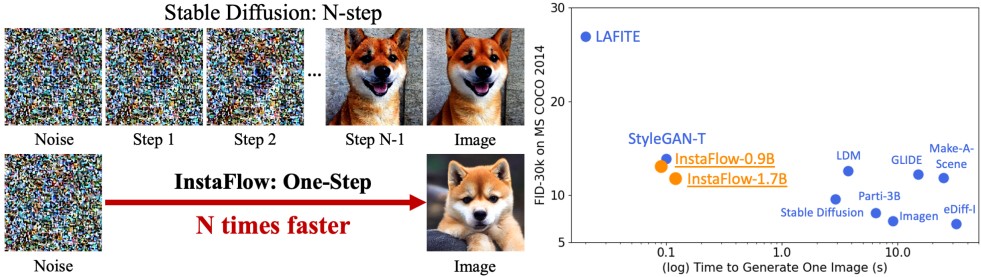

Figure 1: InstaFlow is a high-quality one-step text-to-image model derived from Stable Diffusion [70]. Within 0.1 second, it generates images with similar FID as StyleGAN-T [73] on MS COCO 2014. The whole fine-tuning process to yield InstaFlow is pure supervised learning and costs only 199 A100 GPU days.

## 1 Introduction

Modern text-to-image (T2I) generative models, such as DALL-E [66; 67], Imagen [71; 29], Stable Diffusion [70], StyleGAN-T [73], and GigaGAN [32], have demonstrated the remarkable ability to synthesize realistic, artistic, and detailed images based on textual descriptions. These advancements are made possible through the assistance of large-scale datasets [74] and models [32; 66; 70].

---

*This work was done during an internship at Helixon Research

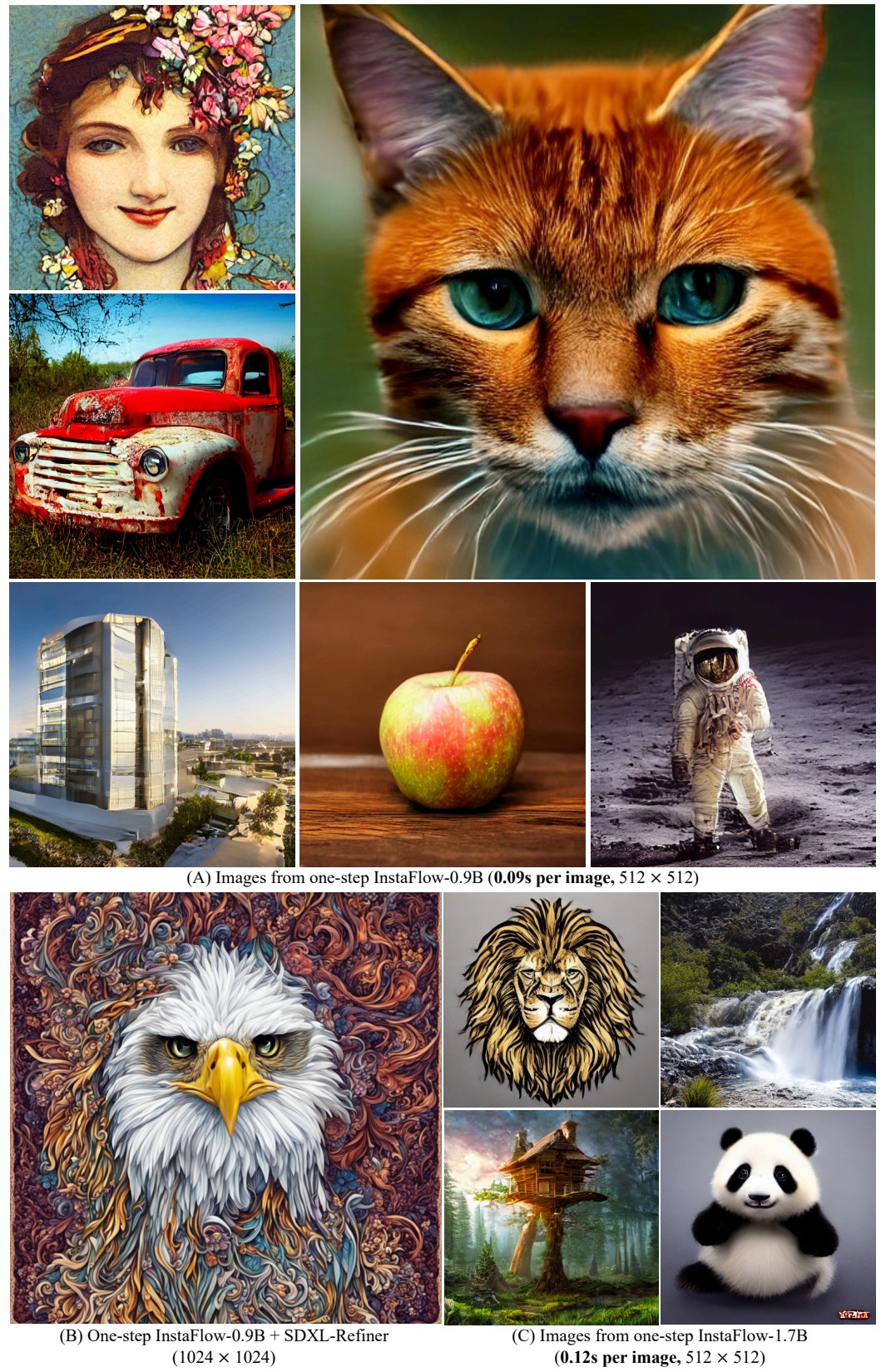

(A) Images from one-step InstaFlow-0.9B (**0.09s per image,** 512 × 512)

(B) One-step InstaFlow-0.9B + SDXL-Refiner
(1024 × 1024)

(C) Images from one-step InstaFlow-1.7B
(**0.12s per image,** 512 × 512)

Figure 2: (A) Examples of 512 × 512 images generated from one-step InstaFlow-0.9B in **0.09s**; (B) The images generated from one-step InstaFlow-0.9B can be further enhanced by SDXL-Refiner [62] to achieve higher resolution and finer details; (C) Examples of 512 × 512 images generated from one-step InstaFlow-1.7B in **0.12s**. Inference time is measured on our machine with NVIDIA A100 GPU.

However, despite their impressive generation quality, these models often suffer from excessive inference time and computational consumption [29; 71; 66; 67; 70]. This can be attributed to the fact that most of these models are either auto-regressive [8; 14; 17] or diffusion models [28; 80]. For instance, Stable Diffusion, even when using a state-of-the-art sampler [41; 49; 77], typically requires more than 20 steps to generate acceptable images. As a result, prior works [72; 58; 82] have proposed employing knowledge distillation on these models to reduce the required sampling steps and accelerate their inference. Unfortunately, these methods struggle in the small step regime. In particular, one-step large-scale diffusion models have not yet been developed. The existing one-step large-scale T2I generative models are StyleGAN-T [73] and GigaGAN [32], which rely on generative adversarial training and require careful tuning of both the generator and discriminator.

In this paper, we present a novel one-step generative model derived from the open-source Stable Diffusion (SD). We observed that a straightforward distillation of SD leads to complete failure. The primary issue stems from the sub-optimal coupling of noises and images, which significantly hampers the distillation process. To address this challenge, we leverage Rectified Flow [45; 43], a recent approach to generative models and optimal transport that learn straight flow models amendable to fast simulation with few or one Euler steps. Rectified flow starts from matching data distribution with a potentially curved flow model (known as 1-flow in [45]), similar to DDIM [77], probability flow ODEs [80] and other flow-based methods [40; 1; 2]. It then deploys an unique *reflow* procedure to straighten the trajectories of the flows, thereby reducing the transport cost between the noise distribution and the image distribution. This improvement in coupling significantly facilitates the distillation process. In this work, we take large-scale text-to-image models as 1-flow, and focus on straightening them with reflow.

Consequently, we succeeded in training the first one-step SD model capable of generating high-quality images with remarkable details. Quantitatively, our one-step model achieves a state-of-the-art FID score of 23.4 on the MS COCO 2017 dataset (5,000 images) with an inference time of only 0.09 second per image. It outperforms the previous fastest SD model, progressive distillation [58], which achieved an one-step FID of 37.2. For MS COCO 2014 (30,000 images), our one-step model yields an FID of 13.1 in 0.09 second, surpassing one of the recent large-scale text-to-image GANs, StyleGAN-T [73] ($13.9$ in $0.1s$). Notably, this is the first time a distilled one-step SD model performs on par with GAN, with pure supervised learning. Discussion of related works is deferred to Appendix A due to limited space.

## 2 METHODS

### 2.1 EFFICIENT INFERENCE IS NEEDED FOR LARGE-SCALE TEXT-TO-IMAGE GENERATION

Recently, various of diffusion-based text-to-image generators [59; 29; 70] have emerged with unprecedented performance. Among them, Stable Diffusion (SD) [70], an open-sourced model trained on LAION-5B [74], gained widespread popularity from artists and researchers. It is based on latent diffusion model [70], which is a denoising diffusion probabilistic model (DDPM) [28; 80] running in a learned latent space. Because of the recurrent nature of diffusion models, it usually takes more than 100 steps for SD to generate satisfying images. To accelerate the inference, a series of post-hoc samplers have been proposed [41; 49; 77]. By transforming the diffusion model into a marginal-preserving probability flow, these samplers can reduce the necessary inference steps to as few as 20 steps [49]. However, their performance starts to degrade noticeably when the number of inference steps is smaller than 10. For the $\leq$10 step regime, progressive distillation [72; 58] is proposed to compress the needed number of inference steps to 2-4. Yet, it is still an open problem if it is possible to turn large diffusion models, like SD, into an one-step model with satisfying quality.

### 2.2 RECTIFIED FLOW AND REFLOW

Rectified Flow [45; 43] is a unified ODE-based framework for generative modeling and domain transfer. It provides an approach for learning a transport mapping $T$ between two distributions $\pi_0$ and $\pi_1$ on $\mathbb{R}^d$ from their empirical observations. In image generation, $\pi_0$ is usually a standard Gaussian distribution and $\pi_1$ the image distribution.

Rectified Flow learns to transfer $\pi_0$ to $\pi_1$ via an ordinary differential equation (ODE), or flow model

$$\frac{\mathrm{d}Z_t}{\mathrm{d}t} = v(Z_t, t), \quad \text{initialized from } Z_0 \sim \pi_0, \text{ such that } Z_1 \sim \pi_1, \tag{1}$$

where $v \colon \mathbb{R}^d \times [0, 1] \to \mathbb{R}^d$ is a velocity field, learned by minimizing a simple mean square objective:

$$\min_v \mathbb{E}_{(X_0, X_1) \sim \gamma} \left[ \int_0^1 \| \frac{\mathrm{d}}{\mathrm{d}t} X_t - v(X_t, t) \|^2 \, \mathrm{d}t \right], \quad \text{with} \quad X_t = \phi(X_0, X_1, t), \tag{2}$$

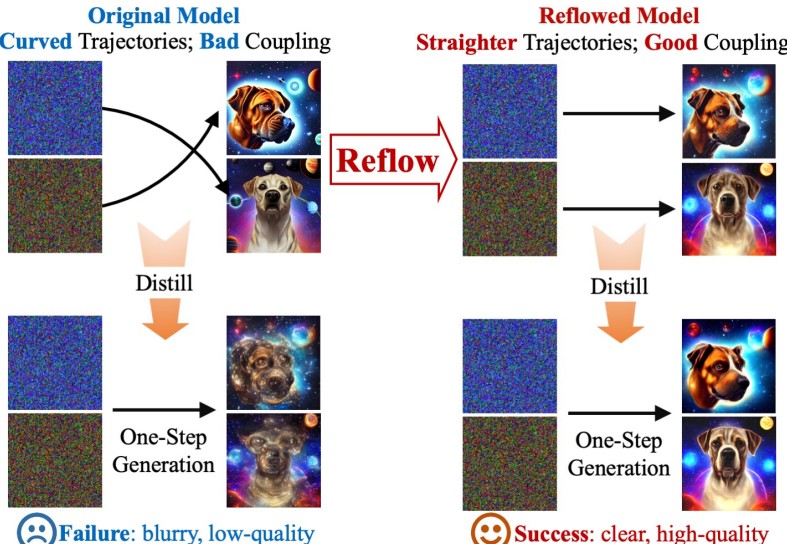

Figure 3: An overview of our pipeline for learning one-step large-scale text-to-image generative models. Direct distillation from pre-trained diffusion models, e.g., Stable Diffusion, fails because their probability flow ODEs have curved trajectories and incur bad coupling between noises and images. After fine-tuned with our text-conditioned reflow, the trajectories are straightened and the coupling is refined, thus the reflowed model is more friendly to distillation. Consequently, the distilled model generates clear, high-quality images in one step. The text prompt is *"A dog head in the universe with planets and stars"*.

where $X_t = \phi(X_0, X_1, t)$ is any time-differentiable interpolation between $X_0$ and $X_1$, with $\frac{\mathrm{d}}{\mathrm{d}t} X_t = \partial_t \phi(X_0, X_1, t)$. The $\gamma$ is any coupling of $(\pi_0, \pi_1)$. A simple example of $\gamma$ is the independent coupling $\gamma = \pi_0 \times \pi_1$, which can be sampled empirically from unpaired observed data from $\pi_0$ and $\pi_1$. Usually, $v$ is parameterized as a deep neural network and equation 2 is solved approximately with stochastic gradient methods. Different specific choices of the interpolation process $X_t$ result in different algorithms. As shown in Liu et al. [45], the commonly used denoising diffusion implicit model (DDIM) [77] and the probability flow ODEs of Song et al. [80] correspond to $X_t = \alpha_t X_0 + \beta_t X_1$, with specific choices of time-differentiable sequences $\alpha_t, \beta_t$ (see Liu et al. [45] for details). In rectified flow, however, the authors suggested a simpler choice of

$$X_t = (1-t)X_0 + tX_1 \qquad \Longrightarrow \qquad \frac{\mathrm{d}}{\mathrm{d}t} X_t = X_1 - X_0, \tag{3}$$

which favors straight trajectories that play a crucial role in fast inference, as we discuss in sequel.

**Straight Flows Yield Fast Generation**   In practice, the ODE in equation 1 need to be approximated by numerical solvers. The most common approach is the forward Euler method, which yields

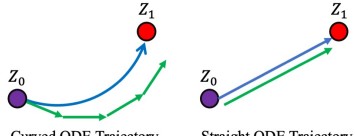

$$Z_{t+\frac{1}{N}} = Z_t + \frac{1}{N} v(Z_t, t), \quad \forall t \in \{0, \ldots, N-1\}/N, \tag{4}$$

Figure 4: ODEs with straight trajectories admits fast simulation.

where we simulate with a step size of $\epsilon = 1/N$ and completes the simulation with $N$ steps. Obviously, the choice $N$ yields a cost-accuracy trade-off: large $N$ approximates the ODE better but causes high computational cost. For fast simulation, it is desirable to learn the ODEs that can be simulated accurately and fast with a small $N$. This leads to ODEs whose trajectory are straight lines. Specifically, we say that an ODE is straight (with uniform speed) if

$$\textit{Straight flow:} \qquad Z_t = tZ_1 + (1-t)Z_0 = Z_0 + tv(Z_0, 0), \quad \forall t \in [0, 1],$$

In this case, Euler method with even a single step ($N = 1$) yields *perfect* simulation; See Figure 4. Hence, straightening the ODE trajectories is an essential way for reducing the inference cost.

**Straightening Text-Conditioned Probability Flows via Text-Conditioned Reflow**   *Reflow* [45] is an iterative procedure to straighten the trajectories of rectified flow without modifying the marginal

---

**Algorithm 1** Training Text-Conditioned Rectified Flow from Stable Diffusion

---

1: **Input:** The pre-trained Stable Diffusion $v_{\texttt{SD}} = v_1$; A dataset of text prompts $D_{\mathcal{T}}$.

2: **for** $k \leq$ a user-defined upper bound **do**
3:    Initialize $v_{k+1}$ from $v_k$.
4:    Train $v_{k+1}$ by minimizing the objective equation 5, where the couplings $(X_0, X_1 = \texttt{ODE}[v_k](X_0 \mid \mathcal{T}))$ can be generated beforehand.

5:    `#NOTE: The trained `$v_k$` is called `$k$`-Rectified Flow.`
6: **end for**

---

**Algorithm 2** Distilling Text-Conditioned $k$-Rectified Flow for One-Step Generation

---

1: **Input:** $k$-Rectified Flow $v_k$; A dataset of text prompts $D_{\mathcal{T}}$; A similarity loss $\mathbb{D}(\cdot, \cdot)$.

2: Initialize $\tilde{v}_k$ from $v_k$.
3: Train $\tilde{v}_k$ by minimizing the objective equation 6, where the couplings $(X_0, X_1 = \texttt{ODE}[v_k](X_0 \mid \mathcal{T}))$ can be generated beforehand.

4: `#NOTE: The trained `$\tilde{v}_k$` is called `$k$`-Rectified Flow+Distill.`

---

distributions, hence allowing fast simulation at inference time. In text-to-image generation, the velocity field $v$ should additionally depend on an input text prompt $\mathcal{T}$ to generate corresponding images. We propose a novel text-conditioned reflow objective,

$$v_{k+1} = \arg\min_v \ \mathbb{E}_{X_0 \sim \pi_0, \mathcal{T} \sim D_{\mathcal{T}}} \left[ \int_0^1 \| (X_1 - X_0) - v(X_t, t \mid \mathcal{T}) \|^2 \, \mathrm{d}t \right],$$

$$\text{with} \ \ X_1 = \texttt{ODE}[v_k](X_0 \mid \mathcal{T}) \ \ \text{and} \ \ X_t = tX_1 + (1-t)X_0,$$

(5)

where $D_{\mathcal{T}}$ is a dataset of text prompts, $\texttt{ODE}[v_k](X_0 \mid \mathcal{T}) = X_0 + \int_0^1 v_k(X_t, t \mid \mathcal{T})\mathrm{d}t$, and $v_{k+1}$ is learned using the same rectified flow objective equation 2, but with the linear interpolation equation 3 of $(X_0, X_1)$ pairs constructed from the previous $\texttt{ODE}[v_k]$.

The key property of reflow is that it preserves the terminal distribution while straightening the particle trajectories and reducing the transport cost of the transport mapping:

1) The distribution of $\texttt{ODE}[v_{k+1}](X_0 \mid \mathcal{T})$ and $\texttt{ODE}[v_k](X_0 \mid \mathcal{T})$ coincides; hence $v_{k+1}$ generates the correct image distribution $\pi_1$ if $v_k$ does so.

2) The trajectories of $\texttt{ODE}[v_{k+1}]$ tend to be straighter than that of $\texttt{ODE}[v_k]$. This suggests that it requires smaller Euler steps $N$ to simulate $\texttt{ODE}[v_{k+1}]$ than $\texttt{ODE}[v_k]$. If $v_k$ is a fixed point of reflow, that is, $v_{k+1} = v_k$, then $\texttt{ODE}[v_k]$ must be exactly straight.

3) $(X_0, \texttt{ODE}[v_{k+1}](X_0 \mid \mathcal{T}))$ forms a better coupling than $(X_0, \texttt{ODE}[v_k](X_0 \mid \mathcal{T}))$ in that it yields lower convex transport costs, that is, $\mathbb{E}[c(\texttt{ODE}[v_{k+1}](X_0 \mid \mathcal{T}) - X_0)] \leq \mathbb{E}[c(\texttt{ODE}[v_k](X_0 \mid \mathcal{T}) - X_0)]$ for all convex functions $c \colon \mathbb{R}^d \to \mathbb{R}$. This suggests that the new coupling might be easier for the student network to learn.

In this paper, we set $v_1$ to be the velocity field of a pre-trained probability flow ODE model (such as that of Stable Diffusion, $v_{\texttt{SD}}$), and denote the following $v_k (k \geq 2)$ as $k$-Rectified Flow.

**Text-Conditioned Distillation** Theoretically, it requires an infinite number of reflow steps equation 5 to obtain ODEs with exactly straight trajectories. However, it is not practical to reflow too many steps due to high computational cost and the accumulation of optimization and statistical error. Fortunately, it was observed in Liu et al. [45] that the trajectories of $\texttt{ODE}[v_k]$ becomes nearly (though not exactly) straight with even one or two steps of reflows. With such approximately straight ODEs, one approach to boost the performance of one-step models is via distillation:

$$\tilde{v}_k = \arg\min_v \mathbb{E}_{X_0 \sim \pi_0, \mathcal{T} \sim D_{\mathcal{T}}} \left[ \mathbb{D}\left( \texttt{ODE}[v_k](X_0 \mid \mathcal{T}), \ X_0 + v(X_0 \mid \mathcal{T}) \right) \right],$$

(6)

where we learn a single Euler step $x + v(x \mid \mathcal{T})$ to compress the mapping from $X_0$ to $\texttt{ODE}[v_k](X_0 \mid \mathcal{T})$ by minimizing a differentiable similarity loss $\mathbb{D}(\cdot, \cdot)$ between images. Learning one-step model with distillation avoids adversarial training [73; 32; 19] or special invertible neural networks [9; 35; 60].

**Distillation and Reflow are Orthogonal Techniques** It is important to note the difference between distillation and reflow: while distillation tries to honestly approximate the mapping from $X_0$ to $\texttt{ODE}[v_k](X_0 \mid \mathcal{T})$, reflow yields a new mapping $\texttt{ODE}[v_{k+1}](X_0 \mid \mathcal{T})$ that can be more regular and smooth due to lower convex transport costs. Reflow is an optional step before distillation, and they are orthogonal to each other. In practice, we find that it is essential to apply reflow to make the mapping $\texttt{ODE}[v_k](X_0 \mid \mathcal{T})$ sufficiently regular and smooth before applying distillation.

**Classifier-Free Guidance Velocity Field for Text-Conditioned Rectified Flow** Classifier-Free Guidance [26] has a substantial impact on the generation quality of SD. Similarly, we propose the following velocity field on the learned text-conditioned Rectified Flow to yield similar effects as Calssifier-Free Guidance,

$$v^{\alpha}(Z_t, t \mid \mathcal{T}) = \alpha v(Z_t, t \mid \mathcal{T}) + (1 - \alpha)v(Z_t, t \mid \texttt{NULL}), \tag{7}$$

where $\alpha$ trades off the sample diversity and generation quality. When $\alpha = 1$, $v^{\alpha}$ reduces back to the original velocity field $v(Z_t, t \mid \mathcal{T})$. We provide analysis on $\alpha$ in Section 4.

## 3 PRELIMINARY RESULTS: REFLOW IS THE KEY TO IMPROVE DISTILLATION

In this section, we conduct experiments with Stable Diffusion 1.4 to examine the effectiveness of the Rectified Flow framework and the reflow procedure. The goal of the experiments in this section is to: 1) examine whether straightforward distillation can be effective for learning a one-step model from pre-trained large-scale T2I prbobility flow ODEs; 2) examine whether text-conditioned reflow can enhance the performance of distillation. Our experiment concludes that: Reflow significantly eases the learning process of distillation, and distillation after reflow successfully produces a one-step model.

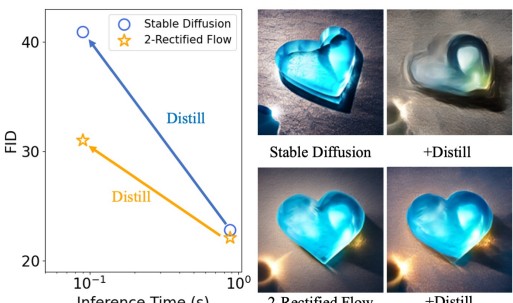

Figure 5: **Left:** The inference time and FID-5k on MS COCO 2017 of all the models. Model distilled from 2-Rectified Flow has a lower FID and smaller gap with the teacher. **Right:** The images generated from different models with the same random noise and text prompt. 2-Rectified Flow refines the coupling between noises and images, making it a better teacher for distillation.

### 3.1 GENERAL EXPERIMENT SETTINGS

In this section, we use the pre-trained Stable Diffusion 1.4 provided in the official open-sourced repository[1] to initialize the weights, since otherwise the convergence is unbearably slow. In our experiment, we set $D_{\mathcal{T}}$ to be a subset of text prompts from laion2B-en [74], pre-processed by the same filtering as SD. $\texttt{ODE}[v_{\text{SD}}]$ is implemented as the pre-trained Stable Diffusion with 25-step DPM-Solver [49] and a fixed guidance scale of 6.0. We set the similarity loss $\mathbb{D}(\cdot, \cdot)$ for distillation to be the LPIPS loss [100]. The neural network structure for both reflow and distillation are kept to the SD U-Net. We use a batch size of 32 and 8 A100 GPUs for training with AdamW optimizer [48]. The choice of optimizer follows the default protocol[2] in HuggingFace for fine-tuning SD.

### 3.2 DIRECT DISTILLATION FAILS, WHILE REFLOW + DISTILLATION SUCCEEDS

**Experiment Protocol** Our investigation starts from directly distilling the velocity field $v_1 = v_{\text{SD}}$ of Stable Diffusion 1.4 with equation 6 without applying any reflow. To achieve the best empirical performance, we conduct grid search on learning rate and weight decay to the limit of our computational resources. For all the models, we train them for $100,000$ steps. We generate $32 \times 100,000 = 3,200,000$ pairs of $(X_0, \texttt{ODE}[v_{\text{SD}}](X_0))$ as the training set for distillation. We compute the Fréchet inception distance (FID) on $5,000$ captions from MS COCO 2017 following the evaluation protocol in [58], then we show the model with the lowest FID in Figure 5. To align the training cost between direct distillation and reflow+distillation for fair comparison, we train 2-Rectified Flow, $v_2$, for $50,000$ steps with the weights initialized from pre-trained SD, then perform

---

[1]https://github.com/CompVis/stable-diffusion
[2]https://huggingface.co/docs/diffusers/training/text2image

| Method | Inf. Time | FID-5k | CLIP |
|---|---|---|---|
| SD 1.4 (25 step)[70] | 0.88s | 22.8 | **0.315** |
| **(Pre) 2-RF (25 step)** | 0.88s | **22.1** | 0.313 |
| PD (1 step)[58] | 0.09s | 37.2 | 0.275 |
| SD 1.4+Distill | 0.09s | 40.9 | 0.255 |
| **(Pre) 2-RF (1 step)** | 0.09s | 68.3 | 0.252 |
| **(Pre) 2-RF+Distill** | 0.09s | **31.0** | **0.285** |

(a) MS COCO 2017

| Method | Inf. Time | FID-30k |
|---|---|---|
| SD* [70] | 2.9s | **9.62** |
| **(Pre) 2-RF (25 step)** | 0.88s | 13.4 |
| SD 1.4+Distill | 0.09s | 34.6 |
| **(Pre) 2-RF+Distill** | 0.09s | **20.0** |

(b) MS COCO 2014

Table 1: Comparison of (a) FID and CLIP score on MS COCO 2017 with $5,000$ images following the evaluation setup in [58] and (b) FID on MS COCO 2014 with $30,000$ images following the evaluation setup in [32]. As in [32; 73], the inference time is measured on NVIDIA A100 GPU with a batch size of 1. 'Pre' is added to distinguish the models from Table 2. 'RF' refers to Rectified Flow; 'PD' refers to Progressive Distillation [58; 72]. $*$ denotes that the numbers are measured by [32].

distillation for another $50,000$ training steps continuing from the obtained $v_2$. To distill from 2-Rectified Flow, we generate $32 \times 50,000 = 1,600,000$ pairs of $(X_0, \texttt{ODE}[v_2](X_0))$ with 25-step Euler solver. The results are also shown in Figure 5 for comparison with direct distillation. The guidance scale $\alpha$ for 2-Rectified Flow is set to $1.5$. For more details, please refer to Appendix.

**Observation and Analysis** We observe that, after $100,000$ training steps, all the models from direct distillation converge. However , as shown in Figure 5, it is difficult for the student model (SD+Distill) to imitate the teacher model (25-step SD), resulting in a huge gap in FID between SD and SD+Distill. On the right side of Figure 5, with the same random noise, SD+Distill generates image with substantial difference from the teacher SD. From the experiments, we conclude that: **direct distillation from SD is a tough learning problem for the one-step student model, which is hard to mitigate by simply tuning the hyperparameters**. In contrast, 2-Rectified Flow refines the coupling between the noise distribution and the image distribution, and eases the learning process for the student model when dis-

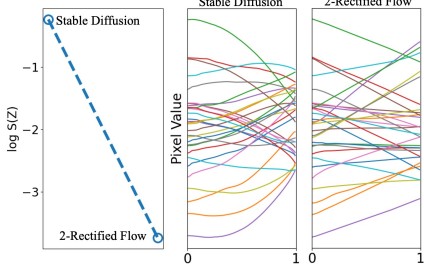

Figure 6: The straightening effect of reflow. **Left:** the straightness $S(Z)$ on different models. **Right:** trajectories of randomly sampled pixels following SD 1.4+DPM-Solver and 2-Rectified Flow.

tillation. It can be inferred from two aspects: (1) The gap between 2-Rectified Flow+Distill and 2-Rectified Flow is much smaller than SD+Distill and SD. (2) On the right side of Figure 5, the image generated from 2-Rectified Flow+Distill shares great resemblance with the original generation, showing that it is easier for the student to imitate. This illustrates that **2-Rectified Flow is a better teacher model to distill an one-step student model than the original SD**.

**Training Cost** Because our 2-Rectified Flow+Distill is fine-tuned from the publicly available pre-trained SD, training only costs $\approx 24.65$ A100 GPU days, which is negligible compared with other large-scale text-to-image models. For reference, the training cost for SD 1.4 from scratch is 6250 A100 GPU days [70]; StyleGAN-T is 1792 A100 GPU days [73]; GigaGAN is 4783 A100 GPU days [32]. A lower-bound estimation of training the one-step SD in Progressive Distillation is 108.8 A100 GPU days [58]. More details can be found in the Appendix.

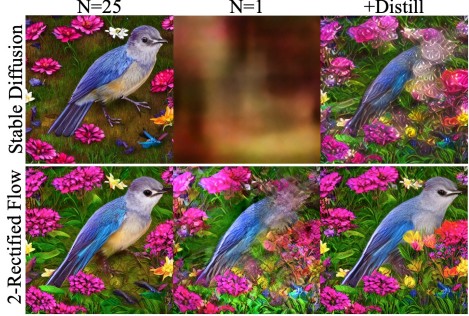

Figure 7: Visual comparison between SD and 2-Rectified Flow. $N$ is number of inference steps.

**Comparison on MS COCO** As shown in Table 1 (a), on MS COCO 2017-5k, (Pre) 2-Rectified Flow can generate realistic images that yield similar FID with SD 1.4 (+DPMSolver [50]) using 25 steps ($22.1 \leftrightarrow 22.8$). Within 0.09s, (Pre) 2-Rectified Flow+Distill gets an FID of 31.0, surpassing the previous best one-step SD model (FID=37.2) distilled from Progressive Distillation [58] with much less training cost (the numbers for Progressive Distillation are measured from Figure 10 in [58] since the model is not publicly available). On MS COCO 2014-30k, (Pre) 2-Rectified Flow+Distill has noticeable advantage (FID=20.0) compared with direct distillation SD 1.4+Distill (FID=34.6) even when (Pre) 2-Rectified Flow has worse performance than the original SD due to insufficient training, indicating the effectiveness of the *reflow* operation.

| Method | Inf. Time | FID-5k | CLIP |
|---|---|---|---|
| SD 1.5 (25 step)[70] | 0.88 | **20.1** | **0.318** |
| **2-RF (25 step)** | 0.88 | 21.5 | 0.315 |
| PD-SD (1 step)[58] | 0.09 | 37.2 | 0.275 |
| **2-RF (1 step)** | 0.09 | 47.0 | 0.271 |
| **InstaFlow-0.9B** | 0.09 | **23.4** | **0.304** |
| PD-SD (2 step)[58] | 0.13 | 26.0 | 0.297 |
| PD-SD (4 step)[58] | 0.21 | 26.4 | 0.300 |
| **2-RF (2 step)** | 0.13 | 31.3 | 0.296 |
| **InstaFlow-1.7B** | 0.12 | **22.4** | **0.309** |

(a) MS COCO 2017

| Cat. | Res. | Method | Inference Time | # Param. | FID-30k |
|---|---|---|---|---|---|
| AR | 256 | DALLE [66] | - | 12B | 27.5 |
| AR | 256 | Parti-750M [96] | - | 750M | 10.71 |
| AR | 256 | Parti-3B [96] | 6.4s | 3B | 8.10 |
| AR | 256 | Parti-20B [96] | - | 20B | 7.23 |
| AR | 256 | Make-A-Scene [18] | 25.0s | - | 11.84 |
| Diff | 256 | GLIDE [59] | 15.0s | 5B | 12.24 |
| Diff | 256 | LDM [70] | 3.7s | 0.27B | 12.63 |
| Diff | 256 | DALLE 2 [67] | - | 5.5B | 10.39 |
| Diff | 256 | Imagen [29] | 9.1s | 3B | 7.27 |
| Diff | 256 | eDiff-I [3] | 32.0s | 9B | 6.95 |
| GAN | 256 | LAFITE [103] | 0.02s | 75M | 26.94 |
| - | 512 | Muse-3B [7] | 1.3s | 0.5B | 7.88 |
| GAN | 512 | StyleGAN-T [73] | 0.10s | 1B | 13.90 |
| GAN | 512 | GigaGAN [32] | 0.13s | 1B | 9.09 |
| Diff | 512 | SD* [70] | 2.9s | 0.9B | 9.62 |
| - | 512 | **2-RF (25 step)** | 0.88s | 0.9B | 11.08 |
| - | 512 | **InstaFlow-0.9B** | 0.09s | 0.9B | 13.10 |
| - | 512 | **InstaFlow-1.7B** | 0.12s | 1.7B | 11.83 |

(b) MS COCO 2014

Table 2: Comparison of (a) FID and CLIP score on MS COCO 2017 with $5,000$ images following the evaluation setup in [58] and (b) FID on MS COCO 2014 with $30,000$ images following the evaluation setup in [32]. 'RF' refers to Rectified Flow; 'PD' refers to Progressive Distillation [58; 72]; 'AR' refers to Autoregressive. $*$ denotes that the numbers are measured by [32].

**Straightening Effects of Reflow** We empirically examine the properties of reflow in text-to-image generation. To quantitatively measure the straightness, we use the deviation of the velocity along the trajectory following [45; 43], that is, $S(Z) = \int_{t=0}^{1} \mathbb{E}\left[|| (Z_1 - Z_0) - v(Z_t, t) ||^2\right] \mathrm{d}t$. A smaller $S(Z)$ means straighter trajectories, and when the ODE trajectories are all totally straight, $S(Z) = 0$. In Figure 6, reflow decreases the estimated $S(Z)$, validating the straightening effect of reflow. Moreover, the pixels in SD travel in curved trajectories, while 2-Rectified Flow has much straighter trajectories. In Figure 7, qualitatively, since SD is curved, one-step generation leads to meaningless noises and SD+Distill fails. Thanks to reflow, one-step generation with 2-Rectified Flow shows recognizable images and distillation from it succeeds.

# 4 INSTAFLOW: SCALING UP FOR BETTER ONE-STEP GENERATION

Our preliminary results using SD 1.4 highlight the benefits of incorporating the reflow procedure in distilling one-step diffusion-based models. However, considering that the training process only consumes 24.65 A100 GPU days, there is a potential for further performance enhancement through scaling up. To this end, we extend the training duration with a larger batch size, totaling 199 A100 GPU days. As a result, we achieve InstaFlow, the first one-step SD model capable of generating high-quality images with intricate details in just 0.09 second. Notably, this performance is on par with StyleGAN-T [73], one of the state-of-the-art GANs in the field.

**InstaFlow-0.9B** We switch to Stable Diffusion 1.5, and keep the same $D_{\mathcal{T}}$ as in Section C. The ODE solver sticks to 25-step DPMSolver [49] for $\mathrm{ODE}[v_{\mathrm{SD}}]$. Guidance scale for SD is slightly decreased to $5.0$ because larger guidance scale makes the images generated from 2-Rectified Flow over-saturated. We still generate $1,600,000$ pairs of data for reflow and distillation, respectively. We apply gradient accumulation to expand the batch size. We spend 75.2 A100 GPU days for reflow to get 2-Rectified Flow, then another 108 A100 GPU days for distillation to get 2-Rectified Flow+Distill. The guidance scale $\alpha$ for 2-Rectified Flow is set to $1.5$ during distillation. We name the distilled model InstaFlow-0.9B since U-Net contains $\sim 0.9$B parameters.

**InstaFlow-1.7B** Expanding the model size is a key step in building modern foundation models [70; 62; 6; 5; 16]. To this end, we stack two U-Nets in series, then remove unnecessary modules after a thorough ablation study (see Appendix for details). This gives us a large neural network, termed Stacked U-Net, with 1.7B parameters and an inference time of $0.12$ second. Starting from 2-Rectified Flow obtained in InstaFlow-0.9B, we spend 39.6 A100 GPU days for distillation with Stacked U-Net. More training details of both models can be found in the Appendix.

**Comparison with State-of-the-Arts on MS COCO** We follow the experiment configuration in Seciton C. In Table 2 (a), our InstaFlow-0.9B gets an FID-5k of 23.4 with an inference time of $0.09s$, which is significantly lower than the previous state-of-the-art, Progressive Distillation-SD (1 step, FID=37.2) with similar distillation cost ($108 \leftrightarrow 108.8$ A100 GPU days). The empirical result indicates that reflow helps improve the coupling between noises and images, and 2-Rectified Flow is an easier teacher model to distill from. By increasing the model size, InstaFlow-1.7B leads to a lower FID-5k of 22.4 with an inference time of $0.12s$. On MS COCO 2014, our InstaFlow-0.9B obtains an FID-30k of 13.10 within $0.09s$, surpassing StyleGAN-T [73] (13.90 in $0.1s$).

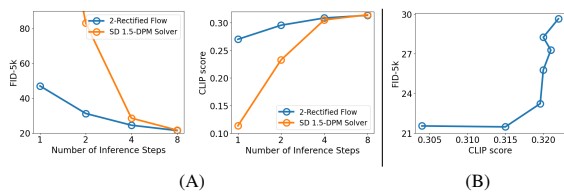

Figure 8: One-step generation from InstaFlow-0.9B. **Left:** With the same random noise, the pose and lighting are preserved across different text prompts. **Right:** Interpolation in the latent space of InstaFlow-0.9B.

**Few-step Generation with 2-Rectified Flow** 2-Rectified Flow has straighter trajectories, which gives it the capacity to generate with extremely few inference steps. We compare 2-Rectified Flow with SD 1.5-DPM Solver [50] on MS COCO 2017. The inference steps are set to $\{1, 2, 4, 8\}$. Figure 9 (A) clearly shows the advantage of 2-Rectified Flow when the number of inference steps $\leq 4$.

Figure 9: (A) Comparison between SD 1.5-DPM Solver and 2-Rectified Flow (with standard Euler solver) in few-step inference. (B) The trade-off curve of applying different $\alpha$ as the guidance scale for 2-Rectified Flow.

**Guidance Scale $\alpha$** It is widely known that guidance scale $\alpha$ is a important hyper-parameter when using Stable Diffusion [26; 70]. Here, we investigate the influence of the guidance scale $\alpha$ for 2-Rectified Flow, which has straighter ODE trajectories. In Figure 9 (B), $\alpha$ increases from $\{1.0, 1.5, 2.0, 2.5, 3.0, 3.5, 4.0\}$, which raises FID-5k and CLIP score on MS COCO 2017 at the same time. The former metric indicates degradation in image quality and the latter metric indicates enhancement in semantic alignment.

**Fast Preview with One-Step InstaFlow** A potential use case of InstaFlow is to serve as previewers. A fast previewer can accelerate the low-resolution filtering process and provide the user more generation possibilities under the same computational budget. Then a powerful post-processing model can improve the quality and increase the resolution. We verify the idea with SDXL-Refiner [62], a recent model that can refine generated images. The one-step InstaFlows generate $512 \times 512$ images in $\sim 0.1s$, then these images are interpolated to $1024$ and refined by SDXL-Refiner to get high-resolution images. Several examples are shown in Figure 10.

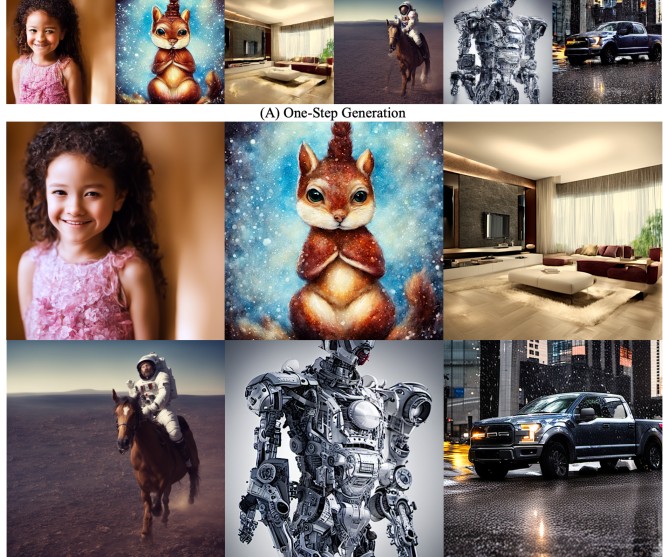

Figure 10: The images generated from our one-step model can be refined by SDXL-Refiner [62] to generate user-preferred high-resolution images on a higher efficiency.

## 5 LIMITATIONS AND CONCLUSIONS

In this paper, we introduce *InstaFlow*, a state-of-the-art one-step text-to-image generator, which is derived from a novel text-conditioned Rectified Flow pipeline with pure supervised learning. Although it may encounter challenges with complex

Figure 11: One of the failure cases.

compositions in the text prompts (see Figure 11 for example), further training with longer duration and larger datasets is likely to mitigate them. InstaFlow significantly closes the gap between continuous-time diffusion models and one-step generative models, inspiring algorithmic innovations and benefiting downstream tasks like 3D generation.

## SOCIETAL IMPACT

This work presents a methodology for accelerating multi-step large-scale text-to-image diffusion models to one-step generators. On a positive note, we believe that the efficiency of these one-step models can lead to energy conservation and environmental benefits, given the extensive utilization of such generative models. Conversely, faster generative models, when manipulated by bad actors, also simplify and speed up the creation of harmful information and fake news. While our work focuses on the scientific insights, these ultra-fast powerful generative models call for advanced techniques through research to ensure their alignment with human values and public interests.

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

# A    RELATED WORKS

**Diffusion Models and Flow-based Models**    Diffusion models [76; 28; 80; 81; 79; 78; 33; 13; 31; 21; 65; 102; 51; 92; 12] have achieved unprecedented results in various generative modeling tasks, including image/video generation [27; 101; 86; 94; 71], audio generation [36], point cloud generation [53; 54; 46; 87], biological generation [91; 55; 86; 30], etc.. Most of the works are based on stochastic differential equations (SDEs), and researchers have explored techniques to transform them into marginal-preserving probability flow ordinary differential equations (ODEs) [80; 77]. Recently, [45; 43; 40; 1; 23] propose to directly learn probability flow ODEs by constructing linear or non-linear interpolations between two distributions. These ODEs obtain comparable performance as diffusion models, but require much fewer inference steps. Among these approaches, Rectified Flow [45; 43] introduces a special *reflow* procedure which enhances the coupling between distributions and squeezes the generative ODE to one-step generation. However, the effectiveness of reflow has only been examined on small datasets like CIFAR10, thus raising questions about its suitability on large-scale models and big data. In this paper, we demonstrate that the Rectified Flow pipeline can indeed enable high-quality one-step generation in large-scale text-to-image diffusion models, hence brings ultra-fast T2I foundation models with pure supervised learning.

**Large-Scale Text-to-Image Generation**    Early research on text-to-image generation focused on small-scale datasets, such as flowers and birds [68; 69; 97]. Later, the field shifted its attention to more complex scenarios, particularly in the MS COCO dataset [39], leading to advancements in training and generation [83; 98; 37]. DALL-E [66] was the pioneering transformer-based model that showcased the amazing zero-shot text-to-image generation capabilities by scaling up the network size and the dataset scale. Subsequently, a series of new methods emerged, including autoregressive models [14; 15; 18; 96], GAN inversion [11; 44], GAN-based approaches [103], and diffusion models [59; 71; 67; 64; 31]. Among them, Stable Diffusion is an open-source text-to-image generator based on latent diffusion models [70]. It is trained on the LAION 5B dataset [74] and achieves the state-of-the-art generalization ability. Additionally, GAN-based models like StyleGAN-T [73] and GigaGAN [32] are trained with adversarial loss to generate high-quality images rapidly. Our work provides a novel approach to yield ultra-fast, one-step, large-scale generative models without the delicate adversarial training.

**Acceleration of Diffusion Models**    Despite the impressive generation quality, diffusion models are known to be slow during inference due to the requirement of multiple iterations to reach the final result. To accelerate inference, there are two categories of algorithms. The first kind focuses on fast post-hoc samplers [33; 42; 49; 50; 77; 4; 10]. These fast samplers can reduce the number of inference steps for pre-trained diffusion models to 20-50 steps. However, relying solely on inference to boost performance has its limitations, necessitating improvements to the model itself [85; 90; 38]. Distillation [25] has been applied to pre-trained diffusion models [52], squeezing the number of inference steps to below 10. Progressive distillation [72] is a specially tailored distillation procedure for diffusion models, and has successfully produced 2/4-step Stable Diffusion [58]. Consistency models [82] are a new family of generative models that naturally operate in a one-step manner. There are several works concurrent with InstaFlow for accelerating large-scale text-to-image models: BOOT [20] designs a data-free distillation pipeline for pre-trained diffusion models with a bootstrapping method; Latent Consistency Model [56] adopts consistency distillation with a skipping scheme; UFOGen [93] combines diffusion models with GAN for high-quality distillation; Yin et al. [95] proposes Distribution Matching Distillation to align the distribution generated from the one-step model with the teacher Stable Diffusion. Instead of employing sophisticated distillation or GAN loss, InstaFlow uses Rectified Flow [45; 43] and its unique *reflow* procedure to straighten the ODE trajectories. With simple supervised learning on a least-squares problem, it refines the coupling between the noise distribution and the image distribution, thereby improving the performance of direct distillation.

# B    NEURAL NETWORK STRUCTURE

The whole pipeline of our text-to-image generative model consists of three parts: the text encoder, the generative model in the latent space, and the decoder. We use the same text encoder and decoder as Stable Diffusion: the text encoder is adopted from CLIP ViT-L/14 and the latent decoder is

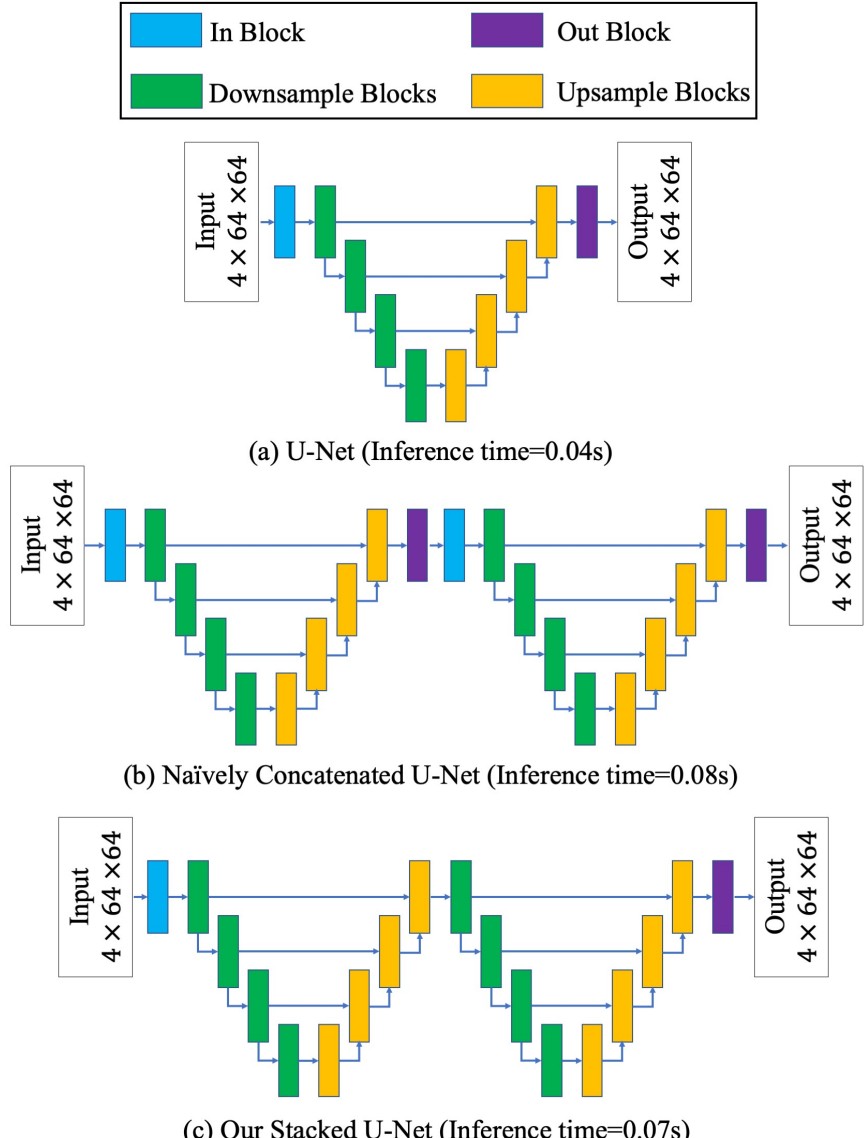

Figure 12: Different neural network structures for distillation and their inference time. The blocks with the same colors can share weights.

adopted from a pre-trained auto-encoder with a downsampling factor of 8. During training, the parameters in the text encoder and the latent decoder are frozen. On average, to generate 1 image on NVIDIA A100 GPU with a batch size of 1, text encoding takes 0.01s and latent decoding takes 0.04s.

By default, the generative model in the latent space is a U-Net structure. For reflow, we do not change any of the structure, but just fine-tune the model. For distillation, we tested three network structures, as shown in Figure 12. The first structure is the original U-Net structure in SD. The second structure is obtained by directly concatenating two U-Nets with shared parameters. We found that the second structure significantly decrease the distillation loss and improve the quality of the generated images after distillation, but it doubles the computational time.

To reduce the computational time, we tested a family of networks structures by deleting different blocks in the second structure. By this, we can examine the importance of different blocks in this concatenated network in distillation, remove the unnecessary ones and thus further decrease inference time. We conducted a series of ablation studies, including:

1. Remove 'Downsample Blocks 1 (the green blocks on the left)'

2. Remove 'Upsample Blocks 1 (the yellow blocks on the left)'

3. Remove 'In+Out Block' in the middle (the blue and purple blocks in the middle).

4. Remove 'Downsample Blocks 2 (the green blocks on the right)'

5. Remove 'Upsample blocks 2 (the yellow blocks on the right)'

The only one that would not hurt performance is Structure 3, and it gives us a 7.7% reduction in inference time ($\frac{0.13-0.12}{0.13} = 0.0769$ ). This third structure, Stacked U-Net, is illustrated in Figure 12 (c).

## C  ADDITIONAL DETAILS AND RESULTS ON THE PRELIMINARY EXPERIMENTS

**General Experiment Settings**   In this section, we use the pre-trained Stable Diffusion 1.4 provided in the official open-sourced repository[3] to initialize the weights, since otherwise the convergence is unbearably slow.

In our experiment, we set $D_{\mathcal{T}}$ to be a subset of text prompts from laion2B-en [74], pre-processed by the same filtering as SD. $\text{ODE}[v_{\text{SD}}]$ is implemented as the pre-trained Stable Diffusion with 25-step DPMSolver [49] and a fixed guidance scale of 6.0. We set the similarity loss $\mathbb{D}(\cdot, \cdot)$ for distillation to be the LPIPS loss [100]. The neural network structure for both reflow and distillation are kept to the SD U-Net. We use a batch size of 32 and 8 A100 GPUs for training with AdamW optimizer [48]. Our training script is based on the official fine-tuning script provided by HuggingFace[4] and the choice of optimizer follows the default protocol. We use exponential moving average with a factor of 0.9999, following the default configuration. We clip the gradient to reach a maximal gradient norm of 1. We warm-up the training process for 1,000 steps in both reflow and distillation. BF16 format is adopted during training to save GPU memory. To compute the LPIPS loss, we used its official 0.1.4 version[5] and its model based on AlexNet. The learning rate for reflow is $10^{-6}$.

We measure the inference time of our models on a server with NVIDIA A100 GPU, and a batch size of 1. We use `PyTorch 2.0.1` and `Hugging Face Diffusers 0.19.3`. For fair comparison, we use the inference time of standard SD on our computational platform for Progressive Distillation-SD as their model is not available publicly. The inference time contains the text encoder and the latent decoder, but does NOT contain NSFW detector.

---

[3]`https://github.com/CompVis/stable-diffusion`
[4]`https://huggingface.co/docs/diffusers/training/text2image`
[5]`https://github.com/richzhang/PerceptualSimilarity`

| FID \ weight decay | $10^{-1}$ | $10^{-2}$ | $10^{-3}$ |
|---|---|---|---|
| learning rate | | | |
| $10^{-5}$ | 44.04 | 45.90 | 44.03 |
| $10^{-6}$ | 137.05 | 134.83 | 139.31 |
| $10^{-7}$ | $\sim$297 | $\sim$297 | $\sim$297 |

Table 3: FID of different distilled SD models measured with 5000 images on MS COCO2017.

## C.1 ADDITIONAL DETAILS AND RESULTS OF DIRECT DISTILLATION

**Grid Search** To achieve the best empirical performance, we conduct grid search on learning rate and weight decay to the limit of our computational resources. Particularly, the learning rates are selected from $\{10^{-5}, 10^{-6}, 10^{-7}\}$ and the weight decay coefficients are selected from $\{10^{-1}, 10^{-2}, 10^{-3}\}$. For all the 9 models, we train them for $100,000$ steps. We generate $32 \times 100,000 = 3,200,000$ pairs of $(X_0, \texttt{ODE}[v_{\text{SD}}](X_0))$ as the training set for distillation.

**Additional Results** We provide additional results on direct distillation of Stable Diffusion 1.4, shown in Figure 18, 19, 20, 21 and Table 3. Although increasing the learning rate boosts the performance, we found that a learning rate of $\geq 10^{-4}$ leads to unstable training and NaN errors. A small learning rate, like $10^{-6}$ and $10^{-7}$, results in slow convergence and blurry generation even after training $100,000$ steps.

## C.2 ADDITIONAL QUANTITATIVE COMPARISON

We provide additional quantitative results with parameter-sharing Stacked U-Net and multiple reflow in Table 4 and 5. According to equation 5, the reflow procedure can be repeated for multiple times. We repeat reflow for one more time to get 3-Rectified Flow ($v_3$), which is initialized from 2-Rectified Flow ($v_2$). 3-Rectified Flow is trained to minimize equation 5 for $50,000$ steps. Then we get its distilled version by generating $1,600,000$ new pairs of $(X_0, \texttt{ODE}[v_3](X_0))$ and distill for another $50,000$ steps. We found that to stabilize the training process of 3-Rectified Flow and its distillation, we have to decrease the learning rate from $10^{-6}$ to $10^{-7}$.

## C.3 ESTIMATION OF TRAINING COST

Measured on our platform, when training with batch size of 4 and U-Net, one A100 GPU day can process $100,000$ iterations using L2 loss, $86,000$ iterations using LPIPS loss; when generating pairs with batch size of 16, one A100 GPU day can generate $200,000$ data pairs. We compute the computational cost according to this.

**Estimated Training Cost of (Pre) 2-Rectified Flow+Distill (U-Net):** $3,200,000/200,000$ (Data Generation) $+ 32/4 \times 50,000/100,000$ (Reflow) $+ 32/4 \times 50,000/86,000$ (Distillation) $\approx 24.65$ A100 GPU days.

**Estimated Training Cost of Progressive Distillation:** We refer to Appendix C.2.1 (LAION-5B $512 \times 512$) of [58] and estimate the training cost. PD starts from 512 steps, and progressively applies distillation to 1 step with a batch size of 512. Quoting the statement 'For stage-two, we train the model with 2000-5000 gradient updates except when the sampling step equals to 1,2, or 4, where we train for 10000-50000 gradient updates', a lower-bound estimation of gradient updates would be 2000 (512 to 256) + 2000 (256 to 128) + 2000 (128 to 64) + 2000 (64 to 32) + 2000 (32 to 16) + 5000 (16 to 8) + 10000 (8 to 4) + 10000 (4 to 2) + 50000 (2 to 1) = 85,000 iterations. Therefore, one-step PD at least requires $512/4 \times 85000/100000 = 108.8$ A100 GPU days. Note that we ignored the computational cost of stage 1 of PD and '2 steps of DDIM with teacher' during PD, meaning that the real training cost is higher than 108.8 A100 GPU days.

| Method | Inference Time | FID-5k | CLIP |
|--------|----------------|--------|------|
| SD 1.4-DPM Solver (25 step)[70; 49] | 0.88s | 22.8 | **0.315** |
| **(Pre) 2-Rectified Flow (25 step)** | 0.88s | **22.1** | 0.313 |
| **(Pre) 3-Rectified Flow (25 step)** | 0.88s | 23.6 | 0.309 |
| Progressive Distillation-SD (1 step)[58] | 0.09s | 37.2 | 0.275 |
| SD 1.4+Distill (U-Net) | 0.09s | 40.9 | 0.255 |
| **(Pre) 2-Rectified Flow (1 step)** | 0.09s | 68.3 | 0.252 |
| **(Pre) 2-Rectified Flow+Distill (U-Net)** | 0.09s | 31.0 | **0.285** |
| **(Pre) 3-Rectified Flow (1 step)** | 0.09s | 37.0 | 0.270 |
| **(Pre) 3-Rectified Flow+Distill (U-Net)** | 0.09s | **29.3** | 0.283 |
| Progressive Distillation-SD (2 step)[58] | 0.13s | 26.0 | 0.297 |
| Progressive Distillation-SD (4 step)[58] | 0.21s | 26.4 | 0.300 |
| SD 1.4+Distill (Stacked U-Net) | 0.12s | 52.0 | 0.269 |
| **(Pre) 2-Rectified Flow+Distill (Stacked U-Net)** | 0.12s | **24.6** | 0.306 |
| **(Pre) 3-Rectified Flow+Distill (Stacked U-Net)** | 0.12s | 26.3 | **0.307** |

Table 4: Comparison of FID on MS COCO 2017 following the evaluation setup in [58]. As in [32; 73], the inference time is measured on NVIDIA A100 GPU, with a batch size of 1, `PyTorch 2.0.1` and `Huggingface Diffusers 0.19.3`. 2-Rectified Flow+Distill outperforms Progressive Distillation within the same inference time using much less training cost. The numbers for Progressive Distillation are measured from Figure 10 in [58]. 'Pre' is added to distinguish the models from Table 2.

| Method | Inference Time | # Param. | FID-30k |
|--------|----------------|----------|---------|
| SD* [70] | 2.9s | 0.9B | **9.62** |
| **(Pre) 2-Rectified Flow (25 step)** | 0.88s | 0.9B | 13.4 |
| SD 1.4+Distill (U-Net) | 0.09s | 0.9B | 34.6 |
| **(Pre) 2-Rectified Flow+Distill (U-Net)** | 0.09s | 0.9B | **20.0** |
| SD 1.4+Distill (Stacked U-Net) | 0.12s | 0.9B | 41.5 |
| **(Pre) 2-Rectified Flow+Distill (Stacked U-Net)** | 0.12s | 0.9B | **13.7** |

Table 5: Comparison of FID on MS COCO 2014 with $30,000$ images. Note that the models distilled after reflow has noticeable advantage compared with direct distillation even when (Pre) 2-Rectified Flow has worse performance than the original SD due to insufficient training. $*$ denotes that the numbers are measured by [32]. 'Pre' is added to distinguish the models from Table 2. As in StyleGAN-T [73] and GigaGAN [32], our generated images are downsampled to $256 \times 256$ before computing FID.

# D   ADDITIONAL TRAINING DETAILS ON INSTAFLOW

**Implementation Details and Training Pipeline for InstaFlow-0.9B**   We switch to Stable Diffusion 1.5, and keep the same $D_{\mathcal{T}}$ as in Section C. The ODE solver sticks to 25-step DPMSolver [49] for `ODE[`$v_{\text{SD}}$`]`. Guidance scale is slightly decreased to $5.0$ because larger guidance scale makes the images generated from 2-Rectified Flow over-saturated. Since distilling from 2-Rectified Flow yields satisfying results, 3-Rectiifed Flow is not trained. We still generate $1,600,000$ pairs of data for reflow and distillation, respectively. To expand the batch size to be larger than $4 \times 8 = 32$, gradient accumulation is applied. The overall training pipeline for 2-Rectified Flow+Distill (U-Net) is summarized as follows:

1. **Reflow (Stage 1)**: We train the model using the reflow objective  equation 5 with a batch size of 64 for 70,000 iterations. The model is initialized from the pre-trained SD 1.5 weights. (11.2 A100 GPU days)

2. **Reflow (Stage 2)**: We continue to train the model using the reflow objective equation 5 with an increased batch size of 1024 for 25,000 iterations. The final model is **2-Rectified Flow**. (64 A100 GPU days)

3. **Distill (Stage 1)**: Starting from the 2-Rectified Flow checkpoint, we fix the time $t = 0$ for the neural network, and fine-tune it using the distillation objective equation 6 with a batch size of 1024 for 21,500 iterations. The guidance scale $\alpha$ of the teacher model, 2-Rectified Flow, is set to $1.5$ and the similarity loss $\mathbb{D}$ is L2 loss. (54.4 A100 GPU days)

4. **Distill (Stage 2)**: We switch the similarity loss $\mathbb{D}$ to LPIPS loss, then we continue to train the model using the distillation objective equation 6 and a batch size of 1024 for another 18,000 iterations. The final model is 2-Rectified Flow+Distill (U-Net). We name it **InstaFlow-0.9B**. (53.6 A100 GPU days)

The total training cost for InstaFlow-0.9B is $3,200,000/200,000$ (Data Generation) $+ 11.2 + 64 + 54.4 + 53.6 = 199.2$ A100 GPU days.

**Implementation Details and Training Pipeline for InstaFlow-1.7B**  We adopt the Stacked U-Net structure in Section C, but abandon the parameter-sharing strategy. This gives us a Stacked U-Net with 1.7B parameters, almost twice as large as the original U-Net. Starting from 2-Rectified Flow, 2-Rectified Flow+Distill (Stacked U-Net) is trained by the following distillation steps:

1. **Distill (Stage 1)**: The Stacked U-Net is initialized from the weights in the 2-Rectified Flow checkpoint. Then we fix the time $t = 0$ for the neural network, and fine-tune it using the distillation objective equation 6 with a batch size of 64 for 110,000 iterations. The similarity loss $\mathbb{D}$ is L2 loss. (35.2 A100 GPU days)

2. **Distill (Stage 2)**: We switch the similarity loss $\mathbb{D}$ to LPIPS loss, then we continue to train the model using the distillation objective equation 6 and a batch size of 320 for another 2,500 iterations. The final model is 2-Rectified Flow+Distill (Stacked U-Net). We name it **InstaFlow-1.7B**. (4.4 A100 GPU days)

**Discussion 1 (Experiment Observations)**  During training, we made the following observations: (1) the 2-Rectified Flow model did not fully converge and its performance could potentially benefit from even longer training duration; (2) distillation showed faster convergence compared to reflow; (3) the LPIPS loss had an immediate impact on enhancing the visual quality of the distilled one-step model. Based on these observations, we believe that with more computational resources, further improvements can be achieved for the one-step models.

**Discussion 2 (One-Step Stacked U-Net and Two-Step Progressive Distillation)**  Although one-step Stacked U-Net and 2-step progressive distillation (PD) need similar inference time, they have two key differences: (1) 2-step PD additionally minimizes the distillation loss at $t = 0.5$, which may be unnecessary for one-step generation from $t = 0$; (2) by considering the consecutive U-Nets as one model, we are able to examine and remove redundant components from this large neural network, further reducing the inference time by approximately $8\%$ (from $0.13s$ to $0.12s$).

## E  ADDITIONAL QUALITATIVE RESULTS

We provide additional qualitative results in Figure 13, 14, 15, 16, including inspections on the latent spaces of InstaFlow-0.9B and InstaFlow-1.7B, and visual comparison of few-step generation and guidance scale $\alpha$. We also show uncurated images generated from 20 random LAION text prompts with the same random noises for visual comparison. The images from different models are shown in Figure 22, 23, 24, 25.

**Alignment between 2-Rectified Flow and the One-Step Models**  The learned latent spaces of generative models have intriguing properties. By properly exploiting their latent structure, prior works succeeded in image editing [89; 34; 24; 84; 57; 47; 101], semantic control [61; 11; 44], disentangled control direction discovery [22; 63; 88; 75], etc.. In general, the latent spaces of one-step generators, like GANs, are usually easier to analyze and use than the multi-step diffusion models. One advantage of our pipeline is that it gives a multi-step continuous flow and the corresponding one-step models simultaneously. Figure 17 shows that the latent spaces of our distilled one-step models align with 2-Rectified Flow. Therefore, the one-step models can be good surrogates to understand and leverage the latent spaces of continuous flow, since the latter one has higher generation quality.

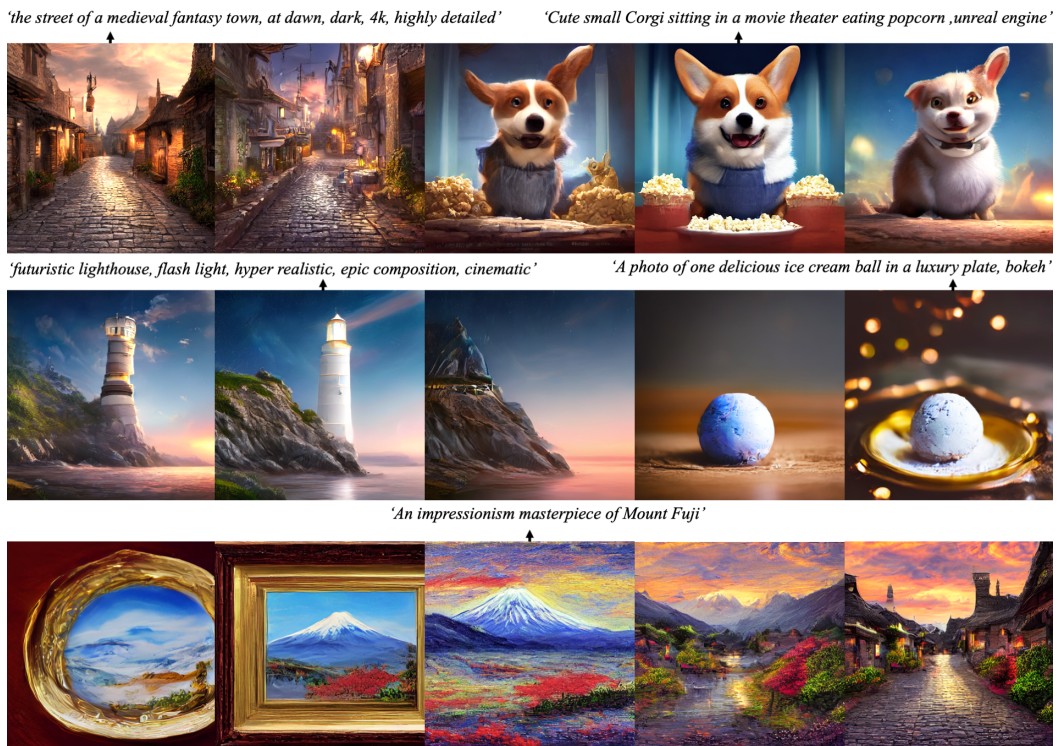

Figure 13: Latent space interpolation of our one-step InstaFlow-0.9B. The images are generated in $0.09s$, saving $\sim 90\%$ of the computational time from the 25-step SD-1.5 teacher model in the inference stage.

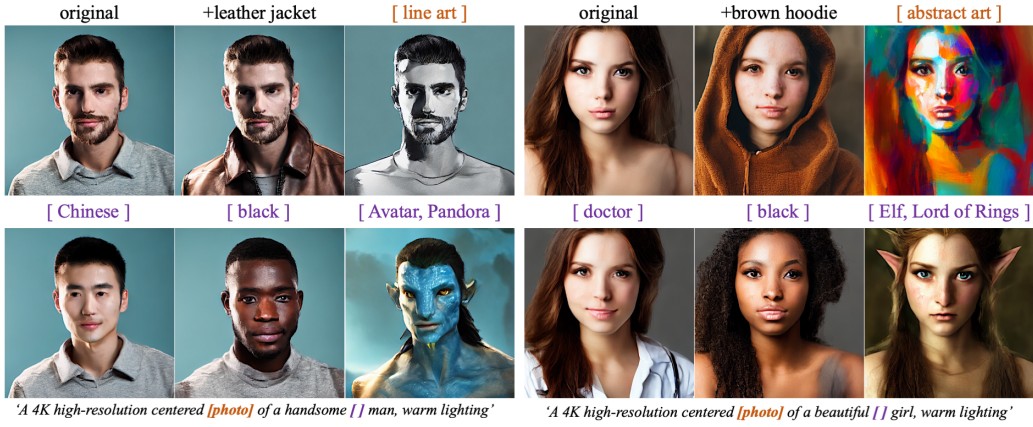

Figure 14: Images generated from our one-step InstaFlow-1.7B in $0.12s$. With the same random noise, the pose and lighting are preserved across different text prompts.

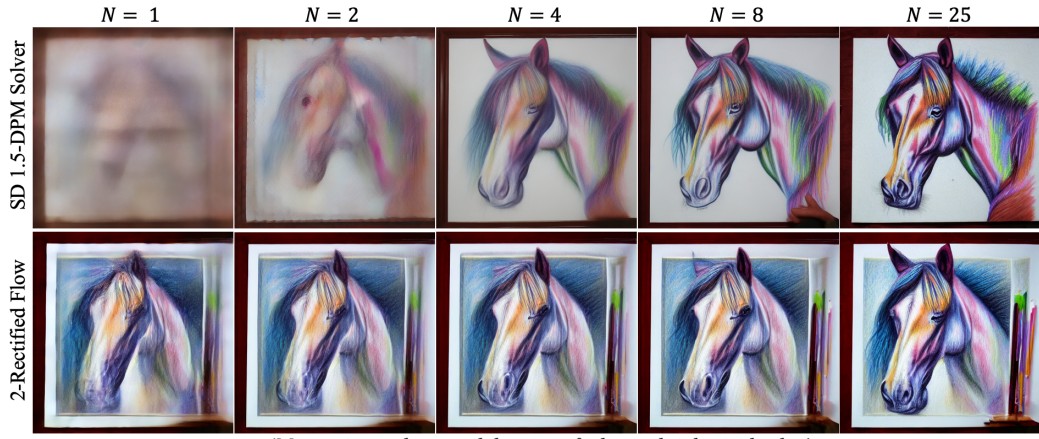

Figure 15: Visual comparison with different number of inference steps $N$. With the same random seed, 2-Rectified Flow can generate clear images when $N \leq 4$, while SD 1.5-DPM Solver cannot.

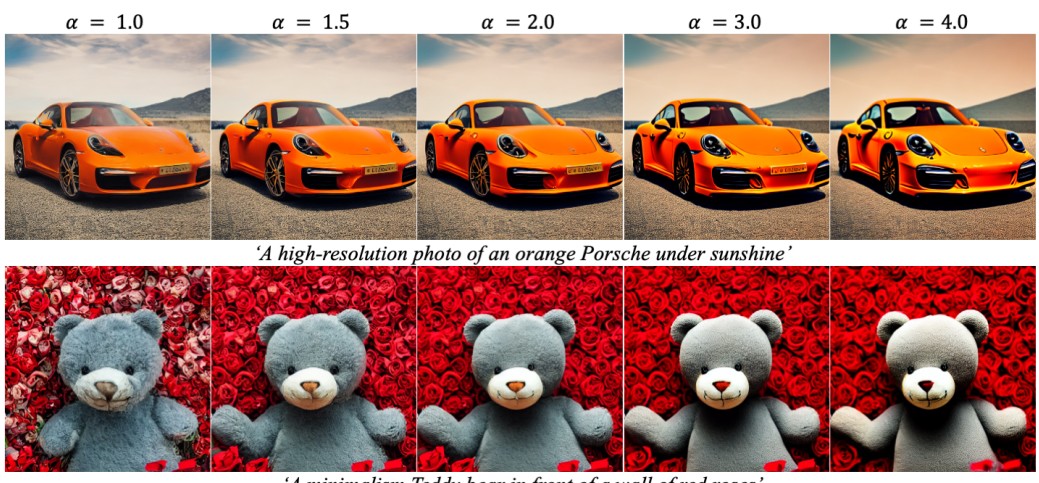

Figure 16: Visual comparison with different guidance scale $\alpha$ on 2-Rectified Flow. When $\alpha = 1.0$, the generated images have blurry edges and twisted details; when $\alpha \geq 2.0$, the generated images gradually gets over-saturated.

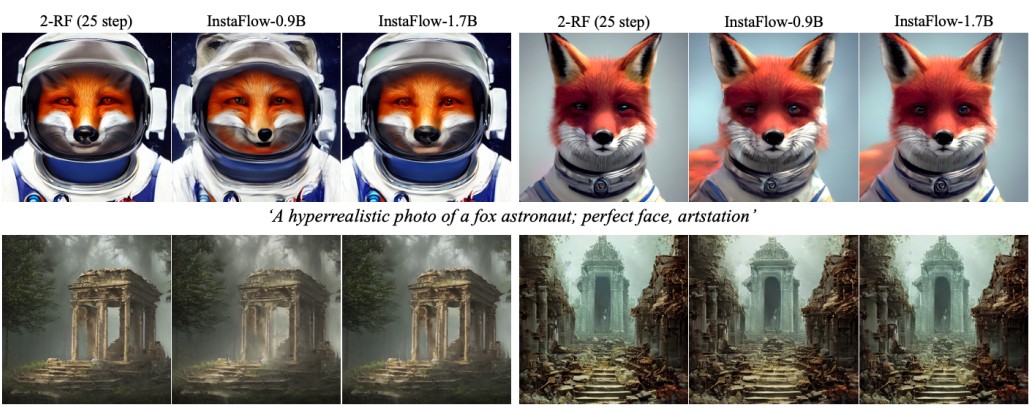

Figure 17: With the same random noise and text prompts, the one-step models generate similar images with the continuous 2-Rectified Flow, indicating their latent space aligns. Therefore, the one-step models can be good surrogates to analyze the properties of the latent space of the continuous flow.

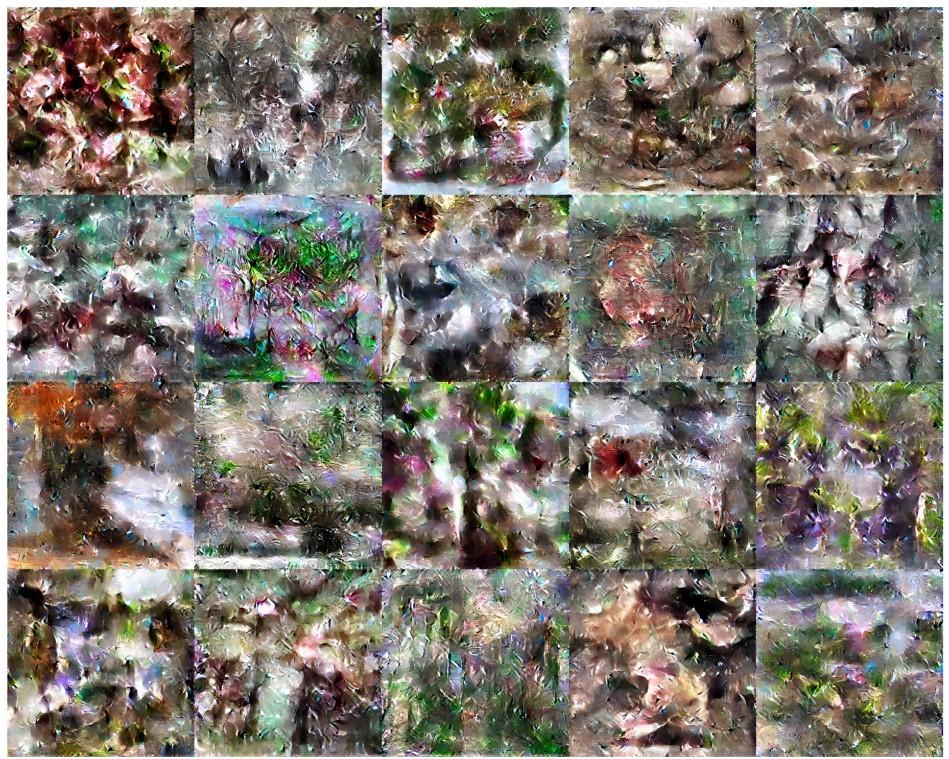

Figure 18: Uncurated samples from SD+Distill (U-Net) trained with a learning rate of $10^{-7}$ and a weight decay coefficient of $10^{-3}$.

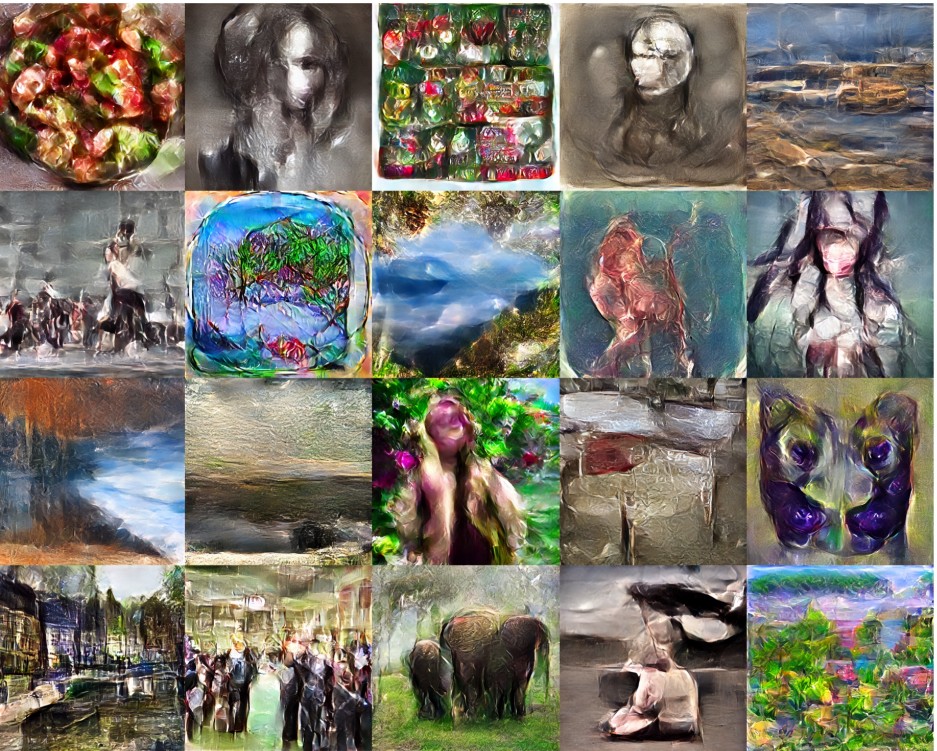

Figure 19: Uncurated samples from SD+Distill (U-Net) trained with a learning rate of $10^{-6}$ and a weight decay coefficient of $10^{-3}$.

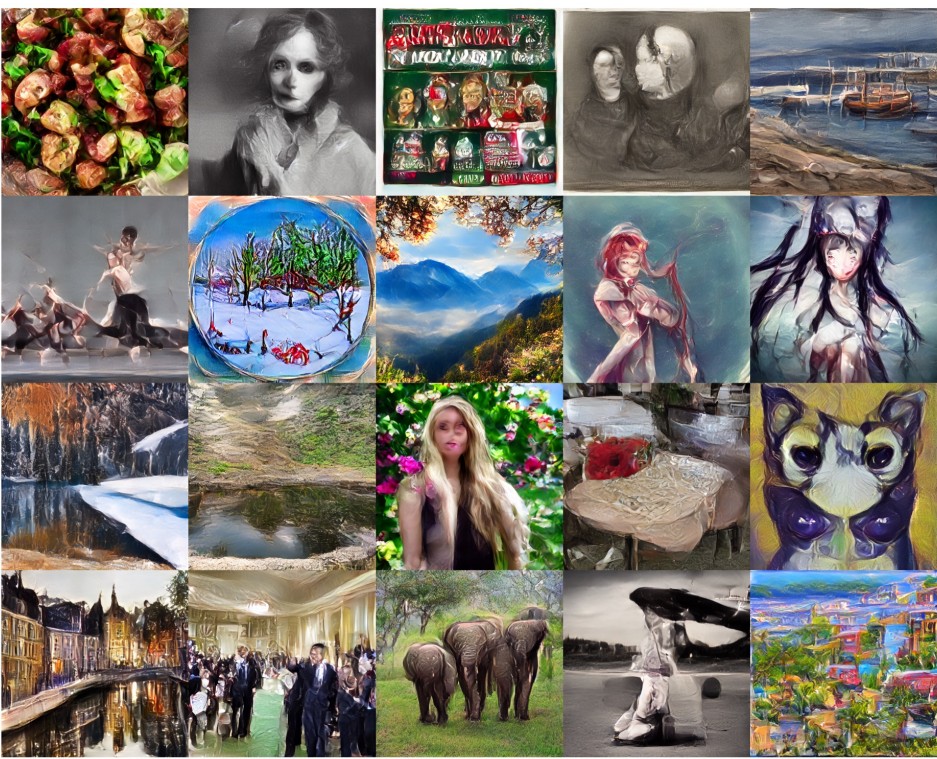

Figure 20: Uncurated samples from SD+Distill (U-Net) trained with a learning rate of $10^{-5}$ and a weight decay coefficient of $10^{-3}$.

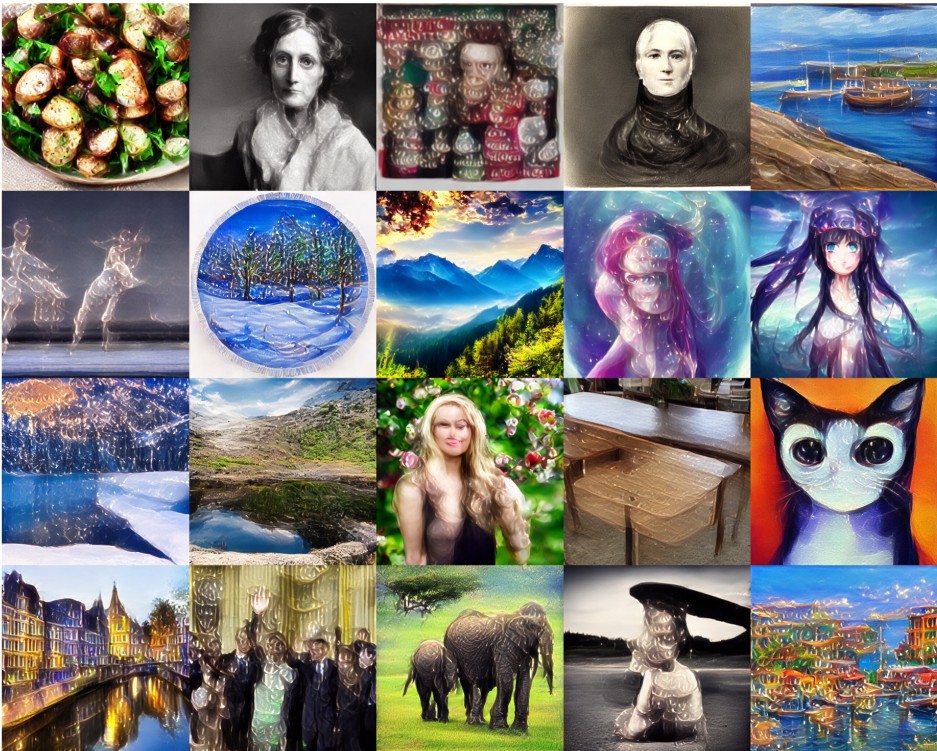

Figure 21: Uncurated samples from SD+Distill (Stacked U-Net) trained with a learning rate of $10^{-5}$ and a weight decay coefficient of $10^{-3}$.

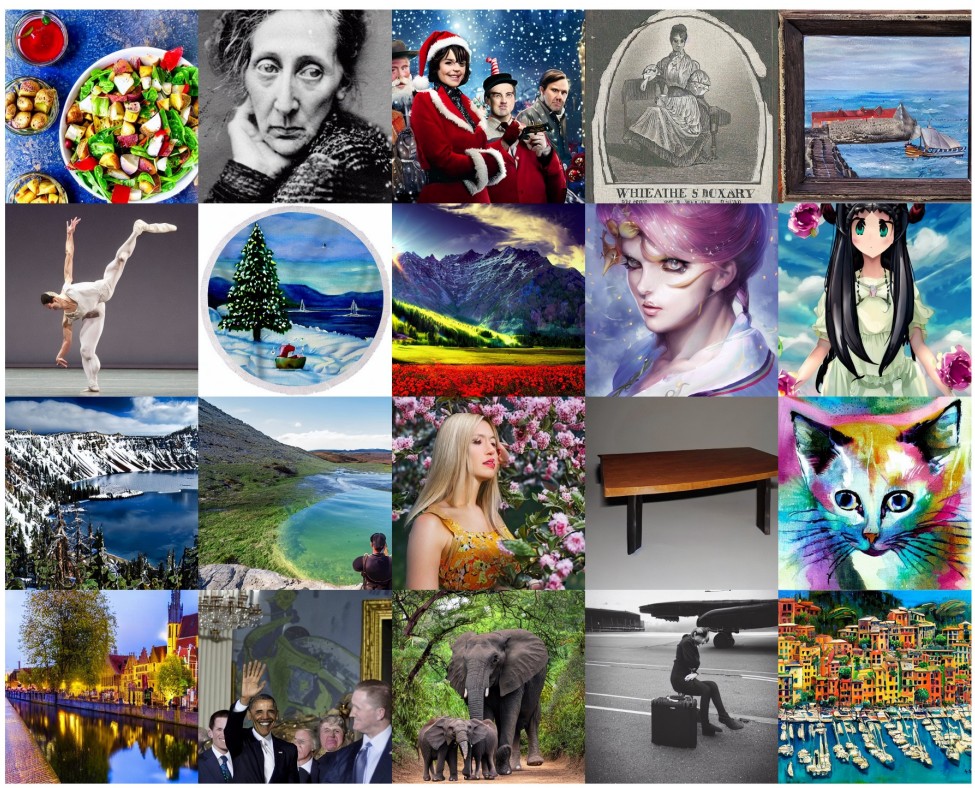

Figure 22: Uncurated samples from Stable Diffusion 1.5 with 25-step DPMSolver [49] and guidance scale 5.0.

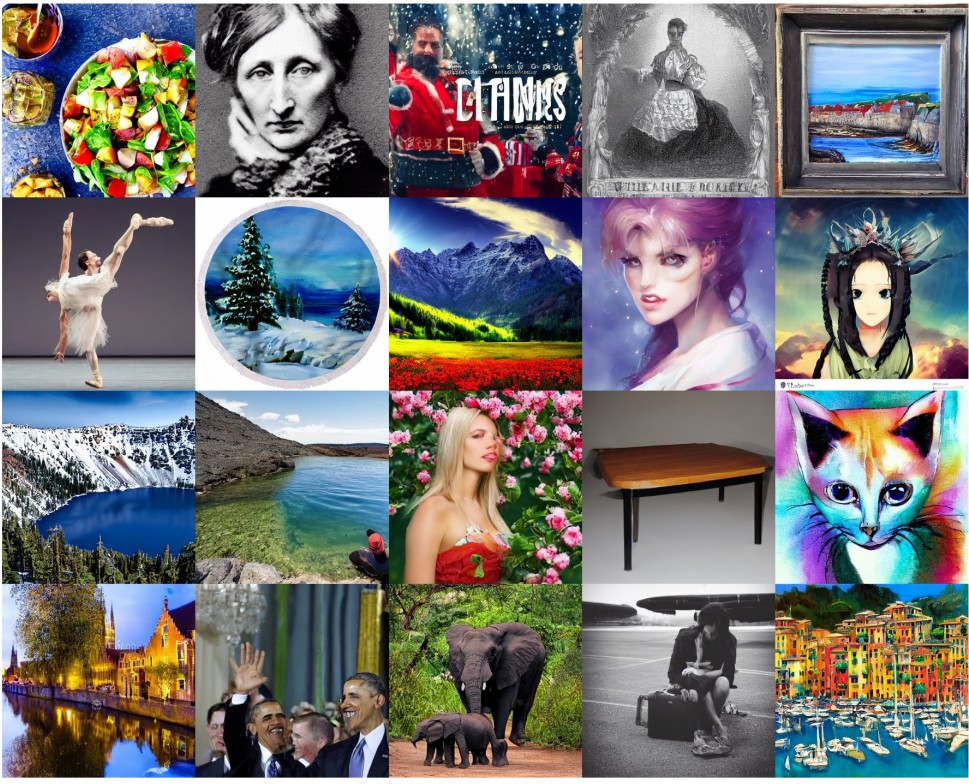

Figure 23: Uncurated samples from 2-Rectified Flow with guidance scale 1.5 and 25-step Euler solver.

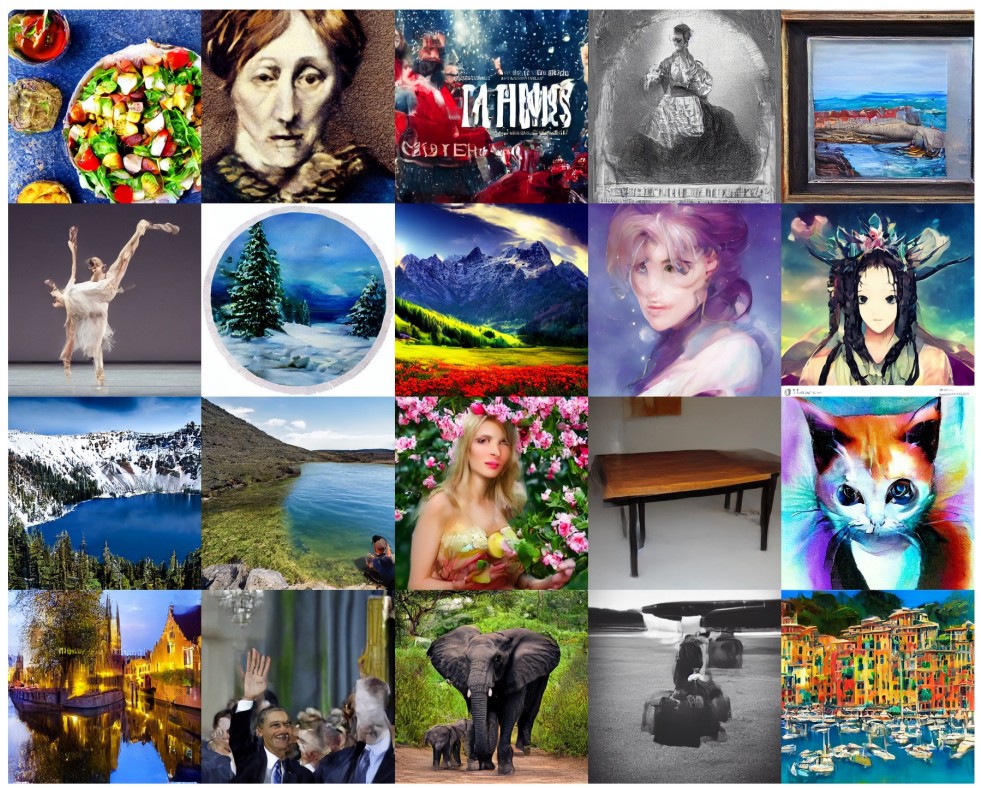

Figure 24: Uncurated samples from one-step InstaFlow-0.9B

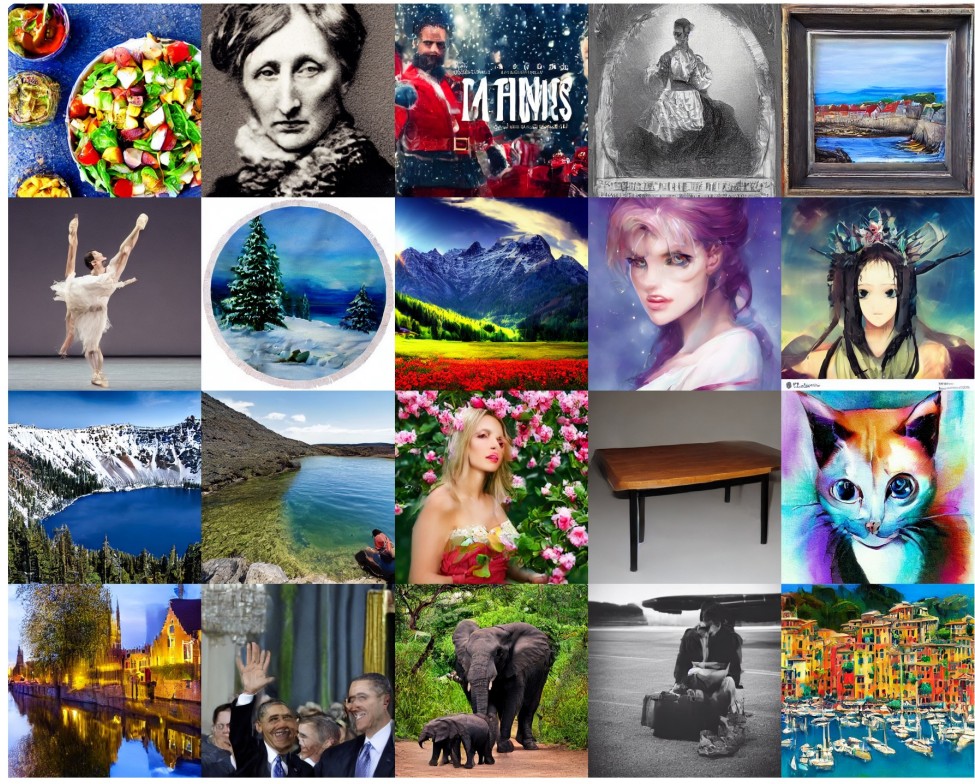

Figure 25: Uncurated samples from one-step InstaFlow-1.7B

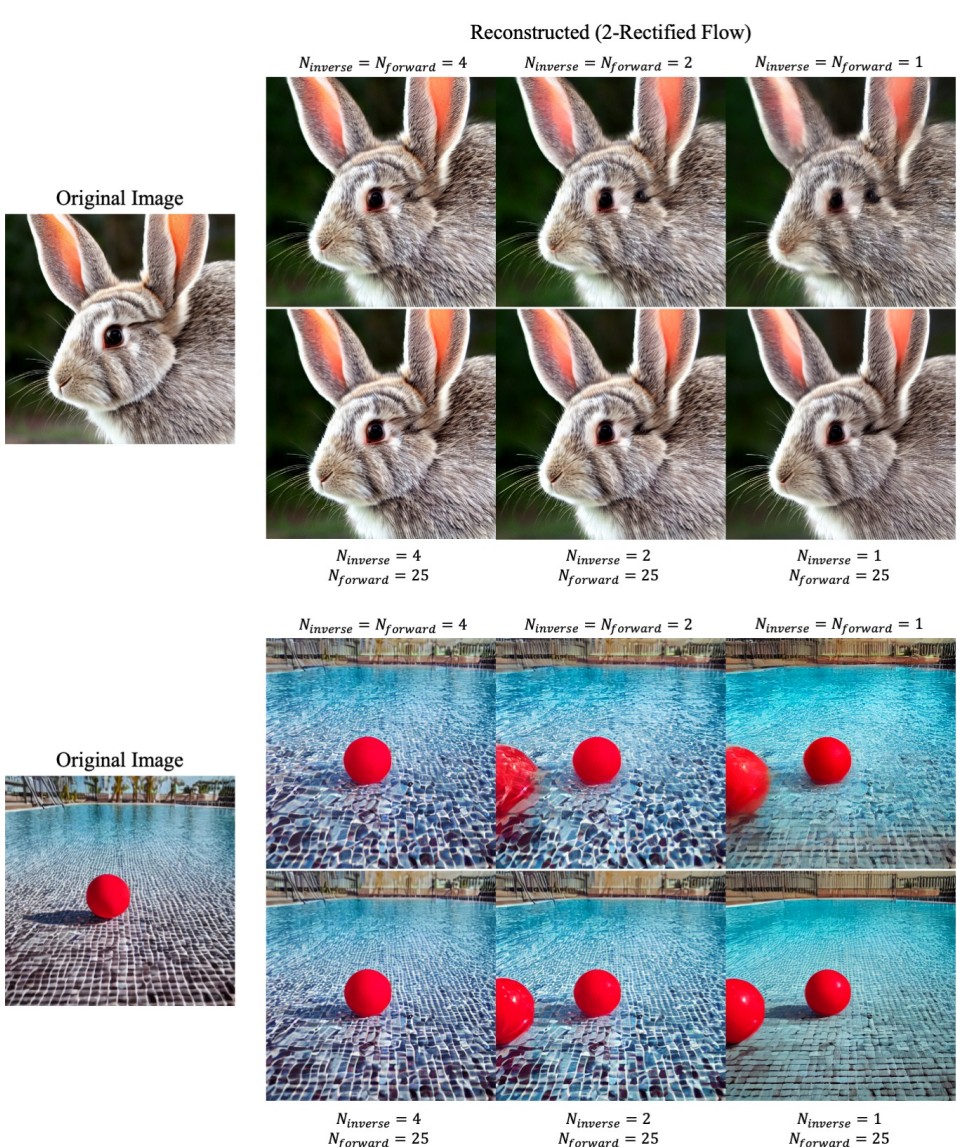

Figure 26: Examples of image encoding and reconstruction with 2-Rectified Flow. Here, we encode an image $X_1$ to the latent noise space by simulating the inverse probability flow ODE, $X_0 = X_1 + \int_1^0 v(X_t, t)\mathrm{d}t$. Then we reconstruct the image from the latent encoding $X_0$ by simulating the forward probability flow ODE, $\hat{X}_1 = X_0 + \int_0^1 v(X_t, t)\mathrm{d}t$. We show the reconstructed image $\hat{X}_1$. For both stages, we adopt Euler solver and use $N_{inverse}$ and $N_{forward}$ steps respectively. With as few as 4 steps, 2-Rectified Flow can encode and reconstruct the original image successfully. Since 2-Rectified Flow is straighter, the encoding stage works well with even 1 step.

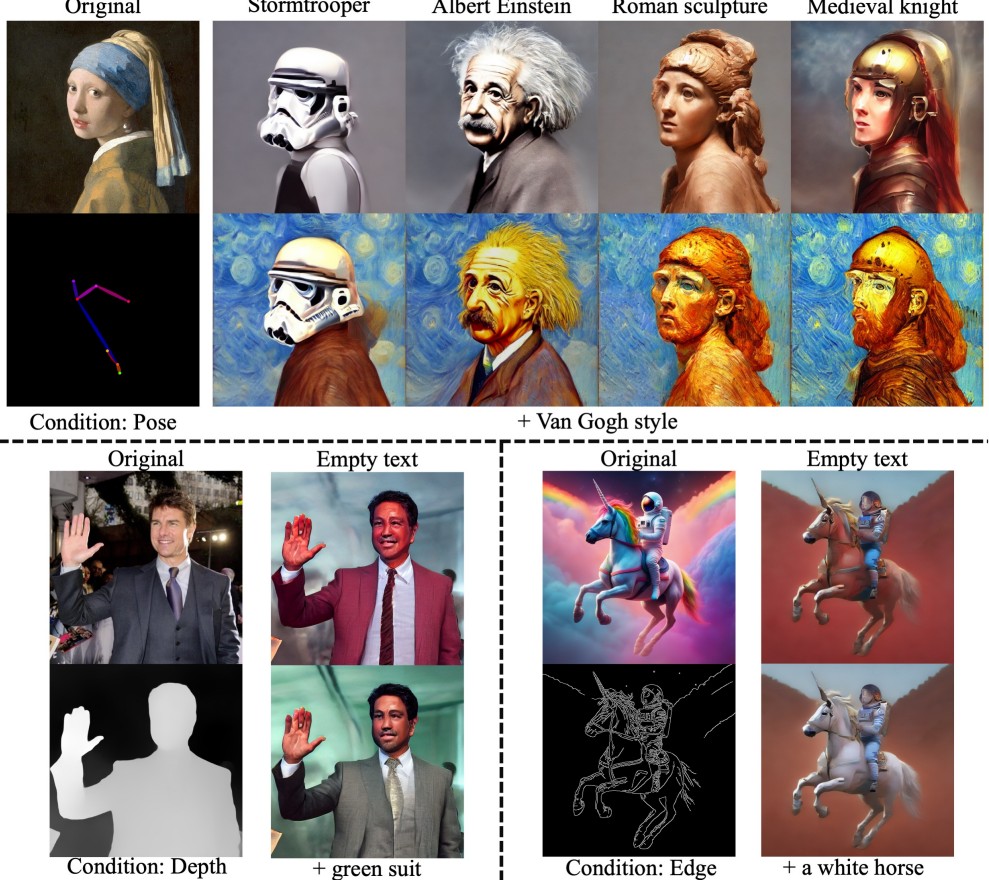

Figure 27: We surprisingly find that pre-trained InstaFlow is fully compatible with ControlNets [99] pre-trained with Stable Diffusion. The images shown here are generated with one-step InstaFlow + pre-trained ControlNets.

