# OpenReview forum: "InstaFlow: One Step is Enough for High-Quality Diffusion-Based Text-to-Image Generation"
_ICLR.cc/2024/Conference — ICLR 2024 poster_

### Official Review · Reviewer_mwbp · 2023-10-29

**Soundness:** 3 good
**Presentation:** 3 good
**Contribution:** 3 good
**Rating:** 8
**Confidence:** 4

**Summary:**

The paper aims to address the limitations of diffusion models in text-to-image (T2I) generation, particularly their slow multi-step sampling process. The authors propose a novel one-step generative model derived from Stable Diffusion (SD) using a method called Rectified Flow. The core of Rectified Flow is its reflow procedure, which improves the coupling between noises and images and facilitates the distillation process.

**Strengths:**

1. Well organized and clarified.
2. The contribution is great. Making the conditional diffusions work with one or very few steps will greatly boost the development of diffusion community.
3. The comparison experiments are carefully and fairly set up, and I appreciate that.

**Weaknesses:**

I have several questions about this paper, and I hope the authors to clearly clarify them.

1. **Storage Overhead**: In my opinion, it seems that either the distillation process or the reflow process actually requires us to create a relatively large (noise, image) pair dataset in advance, which would cause the additional storage overhead.

2. **Intrinsic Difference between the so-called distillation and reflow process**: The distillation step aims to make the model predict the same as target computed by the ODE process of Stable Diffusion at the zero-timestep. While the reflow process seems to only change to make the distillation applied to all possible timesteps.

3. **Noise Scheduler**: The noise scheduler of SD requires a normal diffusion noise scheduler. The reflow requires the "linear" (I call it "linear" just for convenience) scheduler. Wouldn't that cause trouble? Besides, the SD Unet requires time embedding, what time embedding do you use?

4. Why do you choose to predict "x1-x0" instead of "x1-xt"? Do you have any considerations about this?

**Questions:**

I tend to accept the paper, considering its theoretical and technical contributions. However, I have several questions about this paper and hope the authors answer them for me to make the final decision.

---

> ### Author Response · Authors · 2023-11-17
> **Thank you for the review!**
>
> Thank you for the positive review and valuable comments! We will address your concerns below.
>
> **Q#1 (Storage Overhead)**: We admit additional storage space is required to apply our current pipeline. However:
> * Stable Diffusion is trained using more than 1B (text, image) pairs, while InstaFlow only uses 1.6M to achieve the current performance. The overhead is very small.
> * Storage in the training stage is relatively cheap. In contrast, after training, the inference cost of the deployed model usually exceeds the training cost (consider the case of GPT and Midjourney).
> * The distillation step can be further replaced by data-free distillation, like consistency distillation, to save the storage space.
> * Generating the reflow dataset is fast, since only forward inference is involved.
>
> Overall, we argue that the storage overhead is small compared with the original dataset, and the advantage in efficiency brought to the inference stage by our pipeline is significant and overweighs the storage overhead in training.
>
> **Q#2 (Intrinsic Difference between Distillation and Reflow)**: The intrinsic, fundamental difference is the coupling between the noise distribution and the image distribution.
> For distillation, the one-step generative model is represented by,
> $$ f_{distill}(X_0) = X_0 + v_\theta(X_0, 0)$$
>
> For reflow, the continuous-time generative model is represented by,
> $$ f_{flow}(X_0) = X_0 + \int_0^1 v_\theta(X_t, t) d t $$
>
> When distilling from a existing probability flow, the objective is,
> $$ E_{X_0 \sim \pi_0,  X_1 := f_{flow}(X_0)} [D (f_{distill}(X_0), X_1)] $$
> The coupling between noise and image $(X_0, X_1)$ is determined by the flow $f_{flow}$, and the distilled one-step student tries to imitate these couplings. If the couplings $(X_0, X_1)$ are suboptimal, it could be too difficult for the student distilled model to learn, and make the distillation performance bad. Instead, reflow optimizes the whole velocity field, get new probability flow $f_{flow}^{new}$, and refines the coupling $(X_0, X_1^{new}:=f_{flow}^{new}(X_0))$. The new coupling induced from the new flow is easier to distill for the one-step student model. We explained these subtle differences at the top of Page 6 in our submission.
>
> **Q#3 (Noise Scheduler)**: Thank you for the question! We will answer according to our understanding, but further elaboration on the question is welcomed! Rectified flow trains with a new linear interpolated trajectory instead of the old DDPM noise schedule. In our practice, we found the pre-trained Stable Diffusion neural networks have an amazing ability to adapt to the new trajectory without catastrophic forgetting of the learned contents. At the beginning of the project, just like you, we were expecting potential issues, but it turns out the pre-trained networks have superior adaptivity. We think this is a valuable observation to the community as well, since it makes researchers confident to fine-tune with many other alternative trajectories. We will add that to the paper.
>
> **Q#4 (Time Embedding)**:  We use the same time embedding as the original SD. The only difference is that their time steps are integers while we relax that to continuous values.
>
> **Q#5 (Network Prediction)**: We were following the practice in previous works. In fact, predicting $X_1 - X_0$ is equivalent to $X_1 - X_t$ up to a weight coefficient because,
> $$ X_1 - X_t = X_1 - (t X_1 + (1-t)X_0)= (1-t) X_1 - (1-t) X_0 = (1-t) (X_1 - X_0). $$
> Therefore,
> $$ ||X_1 - X_t - f_\theta (X_t, t) ||^2 = || (1-t) (X_1 - X_0) - f_\theta (X_t, t) ||^2 = (1-t)^2|| X_1 - X_0 - \frac{f_\theta (X_t, t)}{1-t}||^2. $$
> So the only difference is the additional time-related weight coefficient $(1-t)^2$ if we parameterize $v_\theta(X_t, t) = \frac{f_\theta (X_t, t)}{1-t}$. The weighting is important as discussed in previous works (e.g., [1]), but we leave that for future investigation as uniform weighting works perfectly in our current practice.
>
> [1] Denoising Diffusion Probabilistic Models, Ho et al.

---

> > ### Comment · Reviewer_mwbp · 2023-11-22
> >
> > Thank you for your feedback.  I will upgrade the rate to accept.
> >
> > Here is some additional advice. Nowadays, many personalized models based on SD are trained and widely used. So I think it's better for the author to train some versatile plug-and-play modules like LoRA that transforms normal diffusion model into instaflow-based diffusion model. This will definitely enlarge the impact of this work.
> >
> > I have seen many reviews in ICLR this year about AIGC. The impact in open-source community makes many papers much easier to get a high rate even with limited novelty.

---

> > > ### Author Response · Authors · 2023-11-22
> > > **Thank you for raising the rating!**
> > >
> > > Thank you for raising the rating!
> > >
> > > We have some internal results that pre-trained ControlNets actually work smoothly with InstaFlow, so we suspect pre-trained LORAs should also work, though we are still implementing. Thank you for your suggestion!

---

### Official Review · Reviewer_W54n · 2023-10-30

**Soundness:** 3 good
**Presentation:** 3 good
**Contribution:** 3 good
**Rating:** 8
**Confidence:** 3

**Summary:**

This paper proposes the InstaFlow, an application that applies the Rectified Flow to Stable Diffusion. The authors implement the rectified flow technique on Stable Diffusion and subsequently distill a one-step diffusion model from the rectified model. The rectified method makes the trajectories of Stable Diffusion straighter, thus making it much easier to distill the multi-step model to fewer or even one-step model.

**Strengths:**

The paper is well-written, and the experiments conducted are both sufficient and convincing. The InstaFlow achieves amazing performance (1-step inference with reasonable quality in approximately 0.09 seconds).

**Weaknesses:**

While this model demonstrates impressive performance, it does involve a trade-off between inference speed and generation quality. From the supplementary document of InstaFlow, we can still observe various artifacts, which may be inherited from the 2-Rectified Flow (e.g., many faces are already distorted in the rectified model). Nevertheless, as the authors also mentioned, this model can be used for generating quick reviews, and then larger models can be employed for further generating high-quality images.

**Questions:**

I don't have further questions regarding the experiments since they're satisfactory to me.
However, given that this work involves practical applications, I encourage the authors to consider sharing the source code and pretrained models, as this would undoubtedly be of great benefit to the community.

---

> ### Author Response · Authors · 2023-11-17
> **Thank you for the review!**
>
> Thank you for the positive review and valuable comments! We will address your concerns below.
>
> **Q#1 (Performance loss)**: We observe the training of 2-Rectified Flow is not totally converged and most text-to-image models are trained using more than 1B text prompts (InstaFlow only uses 1.6M) and thousands of GPU days (InstaFlow only uses 199). Therefore, we believe InstaFlow can be hugely improved by investing more computation and resources, though that is beyond our reach. The emphasis of our research is that text-conditioned reflow has a decisive impact on distilling Stable Diffusion, and we get state-of-the-art one-step Stable Diffusion from that insight.
>
> **Q#2 (Open-source)**: We will open-source the training/inference codes along with pre-trained checkpoints upon acceptance.

---

> > ### Comment · Reviewer_W54n · 2023-11-21
> >
> > Thank you for the feedback. While this paper seems to be an extensive work based on the Rectified Flow, it's not trivial to integrate this technique into the pre-trained Stable Diffusion (SD) and build a fast and practical model by utilizing SD's data prior.
> > After reading other reviewers' comments and authors' responses, I tend to keep my rating.

---

> > > ### Author Response · Authors · 2023-11-21
> > > **Thank you for keeping the positive rating!**
> > >
> > > Thank you for your positive rating and recognition of our contribution!
> > >
> > > Best,
> > >
> > > Authors

---

### Official Review · Reviewer_ZDe9 · 2023-10-31

**Soundness:** 3 good
**Presentation:** 3 good
**Contribution:** 2 fair
**Rating:** 6
**Confidence:** 5

**Summary:**

This paper extends a recently introduced framework of rectified flows to the distillation of the coupling learned by the pretrained diffusion model, e.g. Stable Diffusion (SD).
While the recent work [1] reported the results of experiments on unconditioned generation (on CIFAR-10, LSUN, AFHQ, MetFace and CelebA-HQ datasets) as well as on img2img translation, the current submission focuses on text-conditional generation.
The paper reconfirms that a "rectified" ODE produces an easier target for 1-step distillation.

The main contribution is the InstaFlow model which essentially is a multi-step pipeline which takes a pretrained SD model as an input and outputs a 1-step generative model.
In addition, a novel type of architecture called Stacked U-Net is presented.
As the conducted evaluation shows, InstaFlow outperforms recent baselines such as Progressive Distillation of SD and StyleGAN-T in terms of FID and CLIP score.

[1] Liu et al. Flow Straight and Fast: Learning to Generate and Transfer Data with Rectified Flow. In ICLR, 2023.

**Strengths:**

The writing style of the paper is extremely clear.
It provides a very good introduction to the framework of rectified flows for a reader without deep knowledge of the topic, and overall the manuscript is quite self-contained.
The conducted experiments provide a sufficient support for the motivation of the method. I find the evaluation thorough enough.
The results achieved in the paper are definitely interesting for the broad community of ML researchers and practitioners due to the achieved combination of required computational resources for the model training and inference and its performance.

**Weaknesses:**

1. First of all, this submission is more an extension of the previous work [1] rather than an independent work. The novelty of the presented ideas is definitely limited: all parts of the pipeline were actually introduced previously, and the submitted work applies the same pipeline to the coupling learned by SD model instead of independent coupling of noise and images. The proposed results are definitely valuable for applications. However, they look more like a technical exercise on top of the [1]. Overall, I find this work too incremental although helpful for practitioners.
1. While the idea of Stacked U-Net is interesting, the paper lacks the study if this type of architecture is actually better than increasing the depth (or the number of channels) of the conventional U-Net model.
1. The paper provides the results for latent models only. Taking the empirical nature of this work into account, I suggest adding any of the open cascaded models to the comparison to see, e.g. DeepFloyd IF [2].

[2] https://github.com/deep-floyd/IF

**Questions:**

1. Please, address the limitations discussed above.
1. The training pipeline described in the Appendix D, looks pretty complicated.
    1. Why was it necessary to change the batch size (step 2), and why wasn't more common learning rate tuning applied instead?
    1. What is the reasoning behind switching from L2 to LPIPS objective (step 4) instead of training with a combination of L2 and perceptual loss from the very beginning of the distillation phase?
    1. How were switching points for steps 2 and 4 selected?

---

> ### Author Response · Authors · 2023-11-17
> **Thank you for the review!**
>
> Thank you for the review, comments and valuable questions! We appreciate that the reviewer recognizes the contribution of our paper to the broad ML community. We will address your concerns below. Please kindly raise the rating if our response makes sense to you.
>
> **Q#1 (Novelty)**:  Besides the novelty/difference/interesting phenomenon mentioned in General Response #1, we would like to claim that scaling-up is a resource-consuming process with much back-and-forth, and successful scaling-up is indispensable for the modern machine learning community to acknowledge the value of a specific algorithmic framework. In our work, we use **new large-scale models, new large-scale datasets and new text-conditioned rectified flow framework**, which are not presented in the previous works. Moreover, we extensively provide new empirical comparison and state-of-the-art results on large-scale text-to-image generation, which is a significant improvement over the previous work that has experiments only on toy dataset CIFAR10. The framework and the results, in our opinion, are of great interest to the ICLR community. We will open-source the codes and pre-trained models to further advance the research of generative models.
>
> **Q#2 (Network Structure)**: Thank you for proposing the alternatives! We discuss them below:
> * *Increasing Depth*: Stacked U-Net is actually increasing the depth of the original U-Net by adding more layers. Please refer to Figure 12 c.
> * *Increasing the number of channels*: In our early experiments, we tried to double the number of channels in each layer of the original U-Net, and found no obvious improvement over the original U-Net. This suggests the importance of increasing depth. We will add discussion of this result in the future versions.
>
> **Q#3 (Deep Floyd)**: Thank you for the suggestions! We are definitely interested in applying our text-conditioned rectified flow to more advanced models like DeepFloyd and SDXL. However, for this work, our focus is to fairly compare different distillation methods to provide scientific insights on using text-conditioned reflow before distillation in large-scale text-to-image generation. Limited by the computational resources accessible to us, we leave applications to these advanced models for future development.
>
> **Q#4 (Training Pipeline)**: Thank you for your questions! The pipeline is a little bit complicated because we are limited by resources. While the original Stable Diffusion training adopts 32 * 8 A100 GPUs, we only have 8 A100 GPUs available. Therefore, the training pipeline is a mixture of empirical observations and trade-off between training time and training quality. Even for the original Stable Diffusion, their training pipeline is empirically divided into multiple stages (https://huggingface.co/runwayml/stable-diffusion-v1-5) due to the complexity of training large-scale models. For your questions, we will address the individual concerns below:
> * *Large batch size vs. small learning rate*: We started from training with a batch size of 64, and found the loss converged after around 70,000 iterations (step 1). Then we actually tried both increasing batch size and decreasing learning rate, and found increasing batch size can further reduce the loss while decreasing learning rate has no obvious effect. Increasing batch size is also a common practice when training modern large-scale models, as adopted in GPT-3, PaLM, GLM-130B, etc..
>
> * *Distillation loss*: Optimization on the L2 loss in the latent space of VAE is faster. It also consumes less GPU memory and thus allows larger batch size, because LPIPS loss is computed over the image and the gradient needs to be back-propogated through the VAE decoder. However, optimizing the LPIPS loss has an instant improvement on the generated image quality of the distilled model, so step 4 is indispensable. We actually tried the combination of L2 and LPIPS loss as you suggested, and found the visual quality is worse than using LPIPS loss only. Overall, step 3 is an efficient warm-up stage for step 4 given limited resources.
>
> * *Switching point*: As discussed in sub-question 1, we switch from step 1 to step 2 after observing the (moving average) loss converged with small batch size. In step 2, we train the model up to our computational budget. The loss for training 2-Rectified Flow with a batch size of 1024 is not converged and may improve further given longer training time. We switch from step 3 to step 4 after the quality of the generated images saturates with L2 loss.
>
> If there are enough resources, the training pipeline could be simplified to step 2+step 4 only (large batch size + LPIPS loss for distillation), and the training duration can be expanded for further improvement. The training pipeline could be suboptimal due to resource limitation but will be a good starting point for future research with our open-source.

---

> > ### Comment · Reviewer_ZDe9 · 2023-11-20
> >
> > I would like to thank the authors for their feedback. I encourage the authors to include the discussion on the training strategy (Q#4) to the revised manuscript

---

> > > ### Author Response · Authors · 2023-11-20
> > > **Thank you for the reply!**
> > >
> > > Thank you for the reply!
> > >
> > > We will add the discussions to our later revised versions.
> > >
> > > We are also willing to address any further concerns you may have. If there are no further concerns, we would greatly appreciate your kind consideration in raising your rating.
> > >
> > > Thank you once again!

---

> > > > ### Comment · Reviewer_ZDe9 · 2023-12-01
> > > >
> > > > Having read other reviews, and the authors' feedback,  I still have heavy concerns about the novelty of the work and still think that this is quite a straightforward application of the method known from prior literature. I agree that scaling to large datasets and architectures requires significant engineering resources and a large amount of compute. However, I am not sure if the number of GPU days should be the primary motivation for the decision on the acceptance. Nevertheless, taking into account that the results of the work may be interesting for the practitioners, I tend to raise my the score.

---

### Official Review · Reviewer_VVpt · 2023-11-01

**Soundness:** 4 excellent
**Presentation:** 3 good
**Contribution:** 2 fair
**Rating:** 6
**Confidence:** 4

**Summary:**

This paper successfully demonstrates the use of RECTIFIED FLOW to linearize the model's sampling trajectory, followed by distillation to enhance the sampling speed of the ODE model. It proposes a method for distilling Text-Conditioned flow models, showcasing a variety of ablation studies and results across multiple settings.

**Strengths:**

The results of this paper are truly captivating. It manages to generate images of impressive quality with just one or two steps. The quality showcased in the figures is highly satisfying. In addition, the paper provides a detailed account of various experiments and the corresponding performance metrics, which adds greatly to its value.

**Weaknesses:**

While the paper demonstrates impressive results, it appears to be a straightforward application of RECTIFIED FLOW. I was unable to discern any clear novelty in the algorithms or methods presented. If I'm wrong please kindly let me know the difference.

The model that claims to operate in 1 step actually resembles a configuration of two UNets linked together, and thus feels closer to a 2-step process. Additionally, when the refined model from SDXL is not applied, the results show a noticeable degradation in high-frequency details.

**Questions:**

Given that RECTIFIED FLOW is trained based on its own trajectory, are there any issues that arise from this approach?

Methods like DDIM inversion also seem like they could be applicable in a flow-based context. I am curious about the results in cases where a small number of steps, close to 2, are used.

**Details Of Ethics Concerns:**

Additionally, I would like to highlight the importance of discussing the ethical implications of the presented work in the paper.

---

> ### Author Response · Authors · 2023-11-17
> **Thank you for the review!**
>
> Thank you for the review, comments and valuable questions! We appreciate that the reviewer recognizes the excellent results of our paper. We will address your concerns below. Please kindly raise the rating if our response makes sense to you.
>
> **Q#1(Novelty)**: Besides the novelty/difference/interesting phenomenon mentioned in General Response #1, we would like to claim that scaling-up is never straightforward. Rather, it is a resource-consuming process with much back-and-forth. However, despite the difficulties, successful scaling-up is indispensable for the modern machine learning community to acknowledge the value of a specific algorithmic framework. The **new, large-scale, extensive empirical comparison and state-of-the-art generation results** in our work, using **the novel text-conditioned reflow framework**, sets a solid foundation for the value of the methodologies / theories and inspires practitioners / methodologists to improve the existing algorithms in the future. This, in our opinion, is of great interest to the ICLR community. We will open-source the codes and pre-trained models to further advance the research of generative models.
>
> **Q#2 (Network Architecture)**: Please refer to General response #2. Most of the results are generated by InstaFlow-0.9B using the original U-Net as SD.
>
> **Q#3 (Performance Loss)**: We observe the training of 2-Rectified Flow is not totally converged and most text-to-image models are trained using more than 1B text prompts (InstaFlow only uses 1.6M) and thousands of GPU days (InstaFlow only uses 199).  Therefore, we believe InstaFlow can be hugely improved by investing more computation and resources, though that is beyond our reach. The emphasis of our research is that text-conditioned reflow has a decisive impact on distilling Stable Diffusion, and we get state-of-the-art one-step Stable Diffusion from that insight.
>
> **Q #4 (Flow Trajectory)**: Thank you for the question! We will answer according to our understanding, but further elaboration on the question is welcomed! InstaFlow abandons the original DDPM trajectory of the Stable Diffusion, and re-trains a new linear interpolated trajectory. In our practice, we found the pre-trained Stable Diffusion neural networks have an amazing ability to adapt to the new trajectory without catastrophic forgetting of the learned contents. At the beginning of the project, just like you, we were expecting potential issues, but it turns out the pre-trained networks have superior adaptivity. We think this is a valuable observation to the community as well, since it makes researchers confident to fine-tune with many other alternative trajectories. We will add that to the paper. Thank you for asking again.
>
> **Q #5 (DDIM Inversion)**: Thank you for the suggestion! We put additional results in Figure 26 of the revised Appendix. 2-Rectified Flow can efficiently encode the image to the latent noise space with even 1 step.
>
> **Q #6 (Ethical Implications)**: Thank you for the suggestion! We have added a discussion of societal impact in the revised version of the paper.
>
> Further discussion are welcomed!

---

> > ### Comment · Reviewer_VVpt · 2023-11-19
> >
> > Thank you for authors.
> >
> > I have resolved most of my initial queries after reading the response and revisiting your paper. However, a new question has emerged regarding the classifier-free guidance (CFG) aspect. Authors applied the original CFG formula directly to the velocity field v in its text-guided version. Although this seems plausible to me and appears to not significantly conflict with the existing rectified approach, I'm curious about the underlying intuition. Was this an experimental decision that fortunately yielded positive results, or was there a specific rationale behind it?
> >
> > Additionally, when employing CFG, the approach relies on conditional distributions of both null and context T. Here, I wonder if the marginal property always ensures successful outcomes, or if this is more of an empirical success lacking theoretical backing?
> >
> > In the context of the rectified paper and its discussion on 2-rectified flow, there's an experimental analysis of the CFG parameter alpha. Do you have any additional experimental results or theoretical insights regarding the guidance scale alpha and the recursive rectified flow (k-step)?
> >
> > As a reviewer, I am keen to understand these aspects in greater depth from the authors' perspective.

---

> > > ### Author Response · Authors · 2023-11-20
> > > **Classifier-Free Guidance on InstaFlow**
> > >
> > > Thank you for the reply!
> > >
> > > We provide additional qualitative results in Appendix Figure 16.
> > >
> > > CFG is important for successful text-to-image generation. Without CFG, Stable Diffusion can only generate blurry images that do not align with the text prompts. Therefore, we wonder the influence of CFG on 2-Rectified Flow, and design the CFG velocity field following the practice of SD (Eq. 7). We empirically found it has a similar effect as in the original SD (please see Figure 16). When $\alpha=1.5$, image quality is good, FID is relatively low and CLIP score is relatively high (Figure 9 (B)). Taking everything into account, we choose $\alpha=1.5$.
> > >
> > > The whole success of CFG is more on the empirical side, as already discussed in the original paper [1]. When training 2-Rectified Flow, we randomly set 10% of the text prompts to NULL to enable CFG in the inference stage. In the training stage, our training objective straightens the probability flows for CFG=1.0, but surprisingly, we found it also straightens the flows for CFG > 1.0 empirically (e.g., Figure 15 uses CFG=1.5). Distillation is also improved (over the original SD) simultaneously for different $\alpha$. It is more of a new phenomenon for now, but we set explaining it theoretically as a future direction. We suppose this is probably because CFG is a linear combination of velocities.
> > >
> > > Please kindly raise the rating if our response makes sense to you. Thank you once again!
> > >
> > > [1] CLASSIFIER-FREE DIFFUSION GUIDANCE, Ho and Salimans, https://arxiv.org/pdf/2207.12598.pdf

---

> > > > ### Comment · Reviewer_VVpt · 2023-11-21
> > > >
> > > > I appreciate the effort and contributions of the authors.
> > > >
> > > > I agree with the author's argument that this paper has demonstrated various possibilities that arise when expanding existing techniques to a large scale through empirical experiments. While I'm not entirely certain about the novelty of the algorithm and the training approach, this paper has provided ample experimentation and is considered valuable.
> > > >
> > > > I have revised my rating from 5 to 6.

---

> > > > > ### Author Response · Authors · 2023-11-21
> > > > > **Thank you for raising the rating!**
> > > > >
> > > > > Thank you for your valuable feedback and for revising your rating. We appreciate your recognition of the efforts and contributions made by us.
> > > > >
> > > > > Best,
> > > > > Authors

---

### Author Response · Authors · 2023-11-17
**Thank you for the Reviews!**

Dear reviewers,

We genuinely thank the reviewers for the valuable suggestions, kind comments and overall positive attitude towards our work. In our response, we will first address the common concerns in General Responses, then answer the individual questions from each reviewer. We wish the reviewers to warmly raise the score if our response effectively addresses the concerns. Further discussions are also welcomed.

We have revised our manuscript according to the suggestions from the reviewers, by adding an analysis of societal impact and Figure 26 in the Appendix. We will further revise our manuscript after the discussion period of ICLR.

Best regards,

Authors

---

### Author Response · Authors · 2023-11-17
**General Response #1: Novelty and Difference**

Our work provides **novel, non-trivial** contributions over previous works:

1. We provide a **new text-conditioned reflow+distillation** pipeline. It has noticeable difference from the unconditional pipeline in previous work:


    * **Previous** -- generate **(N noises, N samples)** for the same condition (NULL condition).

    * **Text-conditioned** -- generate **(1 noise, 1 sample) for N different conditions** because there are overwhelming text prompts and we cannot sample multiple pairs for each condition.

    We felt amazing that even one pair for each text prompt straightens the trajectories for all the text prompts and generalizes well to unseen texts, since the original reflow asks for a million pairs of (noise, sample) for the same condition (NULL condition) to learn straight trajectories. We argue that this new conditioned pipeline and the new phenomenon are valuable contributions to the community, which will inspire future research and analysis.

2. In this work, our new text-conditioned framework is applied to **pre-trained DDPM models**. This is different from the previous work, where reflow is only applied to pre-trained rectified flow. The effectiveness of reflow on pre-trained large-scale DDPM empirically expands its application scenario, because there are many more pre-trained DDPMs than pre-trained rectified flow on various areas, e.g., audio / video generation.

3. We present to the ICLR research community **a novel fact that text-conditioned reflow has a significant impact on distillation performance of large-scale Stable Diffusion**, supported by extensive experiments. Furthermore, we propose **a new combination of datasets, models and training details** to successfully scale up the framework to SD-level, while the previous work is only applied to CIFAR10.  Empirically, our new text-conditioned pipeline brings **one-step Stable Diffusion** surpassing existing baselines with pure supervised learning, accompanied by **a new, thorough recipe** for the community to reproduce. We believe it is a significant leap towards more efficient large-scale text-to-image diffusion models and of abundant interest to practitioners in ICLR community, as already agreed by the reviewers.

---

### Author Response · Authors · 2023-11-17
**General Response #2: Network Structure**

1. InstaFlow-0.9B is a one-step text-to-image model using **the same U-Net structure as Stable Diffusion** that generates detailed clear images in 0.09s. With **the same U-Net structure**, our one-step InstaFlow-0.9B surpasses StyleGAN-T in FID on MS COCO 2014. InstaFlow-1.7B is the one-step generative model that uses Stacked U-Net. We will polish our manuscript in later versions to make these configurations very clear since it seems causing confusion to Reviewer VVpt.

2. Moreover, our goal for designing and training Stacked U-Net in the paper is to point out that, by viewing the student model in distillation as an individual network, we obtain **more freedom** in designing the student architecture. Compared to 2-step Progressive Distillation, which keeps the original U-Net in both of the steps and shares parameters, this freedom allows us to thoroughly examine the concatenation of two U-Nets and figure out a more efficient structure.

3. Thanks to the additional freedom in student architecture, we do agree with Reviewer ZDe9 that more extensive search over the architecture space may lead to a better one-step model. However, such search requires designing novel, non-trivial efficient searching algorithms to reduce the redundant computational consumption. We set it as a future direction because the primary argument of the current submission is that text-conditioned reflow is important for learning one-step large-scale text-to-image generative models, and overwhelming contents on architecture search will distract readers.

---

### Author Response · Authors · 2023-11-17
**General Response #3: Research Transparency**

We will open-source the training/inference codes along with pre-trained checkpoints upon acceptance to help the advancement of the field.

---

### Meta-Review · Area_Chair_4bp7 · 2023-12-09

**Metareview:**

This paper presents an extension of rectified flow to latent diffusion models. On the positive side the submission achieves strong results in accelerating diffusion models to one-step samplers. However, the reviewers have criticized the overall novelty of the approach. All the reviewers have rated the paper above the acceptance bar and the ACs are happy to recommend acceptance.

**Justification For Why Not Higher Score:**

The reviewers criticized the overall novelty of the method.

**Justification For Why Not Lower Score:**

The reviewers still found the results presented in the paper substantial and of interest to the community.

---

### Decision · Program_Chairs · 2024-01-16

Accept (poster)